# Robust Multi-object Matching via Iterative Reweighting of the Graph Connection Laplacian

**Yunpeng Shi**[*]      **Shaohan Li**[†]      **Gilad Lerman**[†]
[*]Program in Applied and Computational Mathematics, Princeton University
[†]School of Mathematics, University of Minnesota
yunpengs@princeton.edu, {li000743, lerman}@umn.edu

## Abstract

We propose an efficient and robust iterative solution to the multi-object matching problem. We first clarify serious limitations of current methods as well as the inappropriateness of the standard iteratively reweighted least squares procedure. In view of these limitations, we suggest a novel and more reliable iterative reweighting strategy that incorporates information from higher-order neighborhoods by exploiting the graph connection Laplacian. We provide partial theoretical guarantees and demonstrate the superior performance of our procedure over state-of-the-art methods using both synthetic and real datasets.

## 1 Introduction

The problem of matching multiple objects is crucial in many data-oriented tasks, such as structure from motion (SfM) [21], simultaneous localization and mapping [4], multi-graph matching [28, 32, 33], community detection [1] and solving jigsaw puzzles [15]. One important instance of this problem is multi-image matching, where one is given a set of 2D images, whose 3D scenes include a fixed set of 3D points, and each image contains a set of 2D keypoints that correspond to the set of 3D points. The goal is to recover the correspondences between the keypoints of all images and the fixed 3D points, given measurements of keypoint matches between some pairs of images. Ideally, a keypoint match between two given images aligns pairs of keypoints that describe the same 3D point. In practice, measurements of keypoint matches can be corrupted. A solution of this problem thus requires the design and analysis of methods with provable robustness to corruption.

The multi-object matching problem can be cast as permutation synchronization (PS) [22]. The latter problem assumes a connected graph $G(V,E)$, where each node has a hidden permutation of a fixed size. For example, in image matching it is the correspondence between indices of the keypoints of the image and the 3D points. These hidden permutations determine relative permutations between graph nodes. In the case of image matching, the relative permutations represent the keypoint matches between pairs of images. Permutation synchronization asks to recover the hidden permutations given measurements of the relative permutations.

The measured relative permutations can be highly corrupted. For example, in SfM, the pairwise keypoint matches are commonly derived by SIFT [18] descriptors, whose accuracy is affected by the scene occlusion, change of illumination, viewing distance and perspective. Moreover, repetitive patterns and ambiguous symmetry in common objects of realistic scenarios result in malicious and self-consistent corruption of matches [31].

More work is needed to address such nontrivial practical cases of inaccurate pairwise measurements. Existing guarantees for permutation synchronization often consider a "uniform" corruption model, which does not reflect real scenarios. Uniformity is pursued in two ways: 1) Using the "uniform" Haar distribution on the permutation group to generate corrupted relative permutations; 2) Choosing the corrupted edges in the graph in a uniform manner, such as randomly corrupting an edge with the same probability, while assuming graphs with uniform topology (e.g., generated by the Erdős-Rényi model). Here we try to carefully understand the drawbacks of previous approaches and develop instead a practically efficient method, with partial guarantees, for nonuniform corruption. We find a surprising relationship of our proposed method to Cycle-Edge Message Passing (CEMP) [17]; thus we also clarify CEMP and improve its implementation in our setting.

## 1.1 Relevant Works

The most common methods of PS, which are described in [8, 13, 22], aim to solve a relaxation of a discrete least squares (LS) formulation. Their LS term is an average of squared Frobenius distances between the estimated and measured relative permutations. Due to the use of relaxation, the accuracy of the algorithms of [8, 13, 22] is not competitive and the speed of the algorithm of [8, 13] is rather slow. Two other relaxations of the LS formulation are MatchLift [8, 13] and MatchALS [34]. They have similar drawbacks as above, but their advantage is the applicability to the setting of partial matching, where the number of keypoints vary among objects. MatchALS is faster than MatchLift, whereas MatchLift is theoretically guaranteed under a special probabilistic model. Wang et al. [30] improve the matching accuracy of [34] by incorporating a geometric constraint on the pixel coordinates of keypoints. A tighter approximation algorithm to the non-convex LS formulation is the projected power method (PPM). It was used earlier for solving the problems of angular synchronization [3, 25] and joint alignment [7]. PPM for permutation synchronization was first briefly tested in [7] and later more carefully studied in [14]. All the above LS-based methods are not suitable for nonuniform corruption scenarios.

There are three additional frameworks for solving some types of group synchronization problems [10, 17, 23]. However, [10] only handles Lie groups and [23] only deals with Gaussian noise without outliers, thus neither [10] nor [23] applies to the setting of this paper. Cycle-edge message passing (CEMP) [17] handles all compact groups, in particular, the permutation group, with both small sub-Gaussian noise and adversarial outliers. However, its strong condition for recovery with adversarial outliers can restrict some interesting cases of nonuniform corruption. Furthermore, it does not directly estimate group elements, but corruption levels. The recent Message Passing Least Squares (MPLS) framework [24] aims to resolve the latter problem in practice. However, this framework was only fully developed for the different problem of rotation synchronization.

## 1.2 This Work

These are the main contributions of this work:

- We clarify the serious limitations of the common least squares methods for the PS problem in handling nonuniform corruptions. A rigorous argument appears in the second part of Theorem 5.2. We also clarify why the standard iteratively reweighted least squares procedure is not a good solution for the PS problem.
- We propose (in §4.1) a simple method for estimating the corruption levels in PS. It directly uses the graph connection Laplacian (GCL). We establish the equivalence of this method with the recent CEMP framework [17] (with a properly chosen metric). Unlike CEMP, our procedure is fully vectorized, its computational complexity linearly depends on the cycle size (see supplementary material) and is simpler to explain.
- We establish a new theory for our simple method, and thus also for CEMP, under a special nonuniform setting (see Theorem 5.2). As far as we know, this setting has not been studied before.
- We propose an iteratively reweighted procedure for solving the PS problems, where the weights are obtained by the above simple method. This procedure is similar to MPLS [24], but has some different choices for the group of permutations (as opposed to rotations). We demonstrate the superior performance of our proposed method in comparison to state-of-the-art methods using nontrivial synthetic and real data.

In §2 we mathematically define the PS problem and introduce notation. In §3 we demonstrate the limitations of previous methods. We propose our method in §4 and provide some theoretical guarantees for a special nonuniform scenario in §5. We compare performance with previous methods in §6. At last, §7 concludes this work.

## 2 Preliminaries

We mathematically formulate our underlying problem, introduce notation and review the notion of GCL.

### 2.1 Permutation Synchronization

We formulate the PS problem, while also establishing some notation. For $m \in \mathbb{N}$, we denote by $[m]$ the set $\{1,...,m\}$ and by $S_m$ the permutation group on $m$ elements. For easier presentation, we equivalently represent each element of $S_m$ by an $m \times m$ doubly-stochastic binary matrix (which is orthogonal) and denote the set of these matrices by $\mathscr{P}_m$. In particular, any permutation $\sigma \in S_m$ can be represented as $\boldsymbol{P} \in \mathscr{P}_m$, where $\boldsymbol{P}(i,j) = 1$ if $\sigma(i) = j$ and $\boldsymbol{P}(i,j) = 0$ otherwise. Clearly, $\mathscr{P}_m$ with matrix multiplication is isomorphic to $S_m$. The PS problem is thus formulated as follows: For $n, m \in \mathbb{N}$, assume a graph $G([n], E)$, where each node $i$ is assigned an unknown ground-truth absolute permutation matrix $\boldsymbol{P}_i^* \in \mathscr{P}_m$ (the star superscript emphasizes ground-truth information). These absolute permutations determine the following set of ground-truth relative permutations $\{\boldsymbol{X}_{ij}^*\}_{ij \in E}$, where $\boldsymbol{X}_{ij}^* := \boldsymbol{P}_i^* \boldsymbol{P}_j^{*-1} = \boldsymbol{P}_i^* \boldsymbol{P}_j^{*\mathsf{T}}$. The goal of PS is to recover $\{\boldsymbol{P}_i^*\}_{i \in [n]}$ from measurements $\{\tilde{\boldsymbol{X}}_{ij}\}_{ij \in E}$ of the relative permutations. We use different letters $\boldsymbol{P}$ and $\boldsymbol{X}$ in order to distinguish between

absolute and relative representative permutation matrices. We note that $\boldsymbol{X}_{ji}^* = \boldsymbol{X}_{ij}^{*\intercal}$ and we similarly assume that $\tilde{\boldsymbol{X}}_{ji} = \tilde{\boldsymbol{X}}_{ij}^{\intercal}$ (where either $\tilde{\boldsymbol{X}}_{ji}$ or $\tilde{\boldsymbol{X}}_{ij}^{\intercal}$ are provided). We may thus suppose that $G([n],E)$ is undirected.

An adversarial corruption framework assumes that $\tilde{\boldsymbol{X}}_{ij} = \boldsymbol{X}_{ij}^*$ on good edges $E_g \subset E$ and $\tilde{\boldsymbol{X}}_{ij} \neq \boldsymbol{X}_{ij}^*$ on bad edges $E_b = E \setminus E_g$ [17]. Following this framework and certain assumptions on $E_g$ and $\{\tilde{\boldsymbol{X}}_{ij}\}_{ij \in E_b}$, one may try to prove or disprove exact recovery of $\{\boldsymbol{X}_{ij}^*\}_{ij \in E}$ by a PS algorithm of interest. One can further assume a noise model for $\{\tilde{\boldsymbol{X}}_{ij}\}_{ij \in E_g}$ and quantify the approximate recovery of $\{\boldsymbol{X}_{ij}^*\}_{ij \in E}$.

## 2.2 Further Notation and Conventions

PS solvers are often described using block matrices as follows. Denote by $\tilde{\boldsymbol{X}}$ the block matrix in $\mathbb{R}^{nm \times nm}$ whose $[i,j]$-th block is $\tilde{\boldsymbol{X}}_{ij}$, for $ij \in E$, and zero otherwise. Denote by $\boldsymbol{P} \in \mathbb{R}^{mn \times m}$ the block matrix whose $i$-th block is $\boldsymbol{P}_i \in \mathscr{P}_m$. Let $\mathscr{P}_m^n$ denote the space of block matrices in $\mathbb{R}^{mn \times m}$, whose blocks are in $\mathscr{P}_m$. For a matrix $\boldsymbol{A}$, $\boldsymbol{A}(i,j)$ indicates its $(i,j)$-th element, and for a block matrix $\boldsymbol{B}$, $\boldsymbol{B}[i,j]$ indicates its $[i,j]$-th block. For $\boldsymbol{A}, \boldsymbol{B} \in \mathbb{R}^{k \times l}$, we denote their Frobenius inner product by $\langle \boldsymbol{A}, \boldsymbol{B} \rangle = \mathrm{Tr}(\boldsymbol{A}^\intercal \boldsymbol{B})$. We denote the blockwise inner product of $\boldsymbol{A}, \boldsymbol{B} \in \mathbb{R}^{mk \times ml}$ by $\langle \boldsymbol{A}, \boldsymbol{B} \rangle_{\mathrm{block}} \in \mathbb{R}^{k \times l}$, where its $(i,j)$-th element is $\langle \boldsymbol{A}[i,j], \boldsymbol{B}[i,j] \rangle$.

We use $\boldsymbol{I}_n$, $\boldsymbol{1}_n$ and $\boldsymbol{0}_n$ to represent the $n \times n$ identity, all-one and all-zero matrices, respectively. We also denote the Kronecker product, elementwise multiplication and elementwise division of matrices by $\otimes$, $\odot$ and $\oslash$, respectively. For $i \in [n]$, let $N(i)$, $N_g(i)$ and $N_b(i)$ denote the sets of neighboring nodes of $i$ in $G([n],E)$, $G([n],E_g)$ and $G([n],E_b)$ respectively. We use the shorthand notation w.h.p. to mean with high probability.

## 2.3 The Graph Connection Weight and Laplacian

Throughout the paper we will need to estimate edge weights that express the similarity of the measured and ground-truth relative permutations. We thus assume in this section a weighted graph $G([n],E)$ with arbitrary edge weights $\{w_{ij}\}_{ij \in E}$ and review relevant definitions and notation. We form the weight matrix $\boldsymbol{W} \in \mathbb{R}^{n \times n}$ such that $\boldsymbol{W}(i,j) = w_{ij}$ for $ij \in E$ and $\boldsymbol{W}(i,j) = 0$ otherwise. We recall that the degree of vertex $i \in [n]$ is $d_i = \sum_{j \in N(i)} w_{ij}$ and the degree matrix $\boldsymbol{D} \in \mathbb{R}^{n \times n}$ is diagonal with $\boldsymbol{D}(i,i) = d_i$. The adjacency matrix $\boldsymbol{E}$ is a weight matrix such that $\boldsymbol{E}(i,j) = 1$ for all $ij \in E$. The graph connection weight matrix (GCW), $\boldsymbol{S}$, and the graph connection Laplacian matrix (GCL), $\boldsymbol{L}$ [26], are defined as follows:

$$\boldsymbol{S} = \boldsymbol{W} \otimes \boldsymbol{1}_m \odot \tilde{\boldsymbol{X}} \quad \text{and} \quad \boldsymbol{L} = \boldsymbol{D} \otimes \boldsymbol{I}_m - \boldsymbol{S}. \tag{1}$$

Note that $\boldsymbol{S}_{ij} = w_{ij} \tilde{\boldsymbol{X}}_{ij}$ for all $i$, $j \in [n]$ and $\boldsymbol{L}_{ij} = -w_{ij} \tilde{\boldsymbol{X}}_{ij}$ for $i \neq j \in [n]$, and $\boldsymbol{L}_{ii} = d_i \boldsymbol{I}_m$ for $i \in [n]$. The normalized GCW is respectively defined as $\overline{\boldsymbol{S}} = (\boldsymbol{D}^{-1}\boldsymbol{W}) \otimes \boldsymbol{1}_m \odot \tilde{\boldsymbol{X}}$, so that $\overline{\boldsymbol{S}}_{ij} = w_{ij} \tilde{\boldsymbol{X}}_{ij} / d_i$ for $i$, $j \in [n]$. Throughout the paper we iteratively estimate the graph weight matrix, GCW and normalized GCW and denote their estimated values at iteration $t$ by $\boldsymbol{W}_{(t)}$, $\boldsymbol{S}_{(t)}$ and $\overline{\boldsymbol{S}}_{(t)}$. In practice, we work with the top eigenvectors of the GCW matrix (or the normalized one). Clearly, this is equivalent to using the bottom eigenvectors of the GCL matrix, and we thus use the term GCL when referring to and naming our method.

## 3 Drawbacks of Existing and Possible Solutions

Most established methods for PS, such as [22, 13, 8, 14], are based on least squares optimization. That is, they aim to find the set of absolute permutations whose relative permutations are "closest", in least squares sense, to the measured ones. More specifically, they minimize the following objective function with $q = 2$ (we formulate this problem with general $q > 0$ for future reference):

$$\min_{\{\boldsymbol{P}_i\}_{i \in [n]} \subset \mathscr{P}_m} \sum_{ij \in E} \|\boldsymbol{P}_i \boldsymbol{P}_j^\intercal - \tilde{\boldsymbol{X}}_{ij}\|_F^q. \tag{2}$$

We note that the optimization problem in (2) with $q = 2$ is equivalent to the following one:

$$\max_{\boldsymbol{P} \in \mathscr{P}_m^n} \langle \boldsymbol{P}\boldsymbol{P}^\intercal, \tilde{\boldsymbol{X}} \rangle. \tag{3}$$

Pachauri et al. [22] approximate the solution of (3) by stacking the top $m$ eigenvectors of the block matrix $\tilde{\boldsymbol{X}}$ and then projecting each block of the resulting matrix on $\mathscr{P}_m$ by the Hungarian algorithm [19]. The state-of-the-art method for solving (3) is the projected power method (PPM) [7, 14]. It first initializes $\boldsymbol{P}_{(1)} \in \mathscr{P}_m^n$ following [22], and then iteratively computes $\boldsymbol{P}_{(t+1)}$, for $t \geq 1$, as follows:

$$\boldsymbol{P}_{(t+1)} = \mathrm{Proj}(\tilde{\boldsymbol{X}}\boldsymbol{P}_{(t)}) = \operatorname*{argmin}_{\boldsymbol{P} \in \mathscr{P}_m^n} \|\boldsymbol{P} - \tilde{\boldsymbol{X}}\boldsymbol{P}_{(t)}\|_F^2. \tag{4}$$

The operator Proj is the blockwise projection onto $\mathscr{P}_m$ that is computed by the Hungarian algorithm.

Both PPM [7] and other least squares methods [13, 8, 22] may tolerate uniform corruption. However, applications give rise to nonuniform corruption. For example, in image matching tasks that appear in 3D reconstruction datasets, the images of the same object may come from different sources of different qualities [27]. Matches of low quality images with other images are often erroneous. That is, their neighboring edges in their corresponding graph $G([n],E)$ are more likely to be corrupted. This heterogeneity of images results in nonuniform topological structure of the bad subgraph $G([n],E_b)$. Unfortunately, none of the previous methods can handle well such structure. We later try to quantify such a structure using a special nonuniform model. We also aim to clarify the failure of these methods in handling it (see last part of Theorem 5.2).

In principle, the above problem with PPM and other least squares methods can be addressed by a proper reweighting procedure that focuses only on good edges $ij \in E_g$. A common global weighing method is iteratively reweighted least squares (IRLS). It has been successfully applied for synchronization-type problems with special continuous groups, such as $SO(d)$ synchronization [5, 12, 29] and camera location estimation [11, 20]. However, we claim that common IRLS methods for special synchronization problems with Lie groups do not directly generalize to synchronization problems with discrete groups, such as $S_m$. Indeed, for our setting, standard IRLS aims to solve (2) with $q = 1$, that is, with least absolute deviations. The common hope is that the minimization of least absolute deviations instead of least squares deviations, which corresponds to $q = 2$, is more robust to adversarial corruption. The standard IRLS solution to (2) with $q = 1$ assumes an initial choice of $\{\boldsymbol{P}_{i,(0)}\}_{i \in [n]} \subset \mathscr{P}_m$ and iteratively computes for $t \geq 1$:

$$w_{ij,(t)}^{\text{IRLS}} = 1/\max\{\|\boldsymbol{P}_{i,(t)}\boldsymbol{P}_{j,(t)}^{\mathsf{T}} - \tilde{\boldsymbol{X}}_{ij}\|_F, \delta\} \tag{5}$$

$$\{\boldsymbol{P}_{i,(t+1)}\}_{i \in [n]} = \underset{\{\boldsymbol{P}_i\}_{i \in [n]} \subset \mathscr{P}_m}{\operatorname{argmin}} \sum_{ij \in E} w_{ij,(t)}^{\text{IRLS}} \|\boldsymbol{P}_i \boldsymbol{P}_j^{\mathsf{T}} - \tilde{\boldsymbol{X}}_{ij}\|_F^2, \tag{6}$$

where $\delta$ is a small regularization constant used to avoid a zero denominator. Unlike Lie groups, the discrete nature of permutations may result in zero residuals, $\|\boldsymbol{P}_{i,(t)}\boldsymbol{P}_{j,(t)}^{\mathsf{T}} - \tilde{\boldsymbol{X}}_{ij}\|_F$, on a few edges in the first few iterations of IRLS. These few edges with zero residuals, including the corrupted ones, are extremely overweighed by (5). As a result, most of the "good" information on other edges is ignored; thus, IRLS can produce poor solutions that are even worse than the given corrupted pairwise matches $\tilde{\boldsymbol{X}}$. Moreover, the solution of (6) typically involves relaxation, which may not be tight given poor edge weights. One may use less aggressive reweighting functions for heavy-tailed noise [6]. However, they are not expected to work well in our discrete scenario. One reason is that their weights are updated from the residuals. Since these residuals lie in a discrete finite space with size $m$, there are very limited choices for the weights and the solution of the weighted least squares problem at each iteration can easily get stuck.

## 4 Our Proposed Method

Our idea is to iteratively and alternately estimate weights that emphasize uncorrupted edges and thus emphasize the underlying relative permutations. Similarly to IRLS, at each iteration the estimate of the absolute permutations is improved by solving (6) with the new estimated weights, instead of those computed by (5). Unlike IRLS, each edge weight is no longer determined by information obtained from only two nodes, but by information obtained from cycles containing the two nodes. The information on cycles is easily obtained by direct matrix multiplication. In §4.1 we explain how to initialize such weights. We establish a mathematical proposition that clarifies this simple approach. We also prove the equivalence of our simple method with the more involved CEMP framework [17]. In §4.2, we assume given weights and discuss weighted least squares (WLS) formulations and solutions. Using the ideas of §4.1 and §4.2, we formulate our complete procedure in §4.3.

### 4.1 Weight Initialization

We estimate a good "similarity measure" and use it to estimate good edge weights. We remark that initial good weights that concentrate around the good edges are crucial for our whole procedure. Indeed we use a tight relaxation of a weighted least square formulation at each iteration and wrong weights can have a bad effect on its solution. For each $ij \in E$, we use the following (correlation) affinity, or similarity measure, $a_{ij}^* := \langle \tilde{\boldsymbol{X}}_{ij}, \boldsymbol{X}_{ij}^* \rangle / m \in [0,1]$. We note that $ij \in E_g$ if and only if $a_{ij}^* = 1$. One can choose the edge weight $w_{ij}$ as the estimated $a_{ij}^*$ or an increasing function of it (we clarify our choice below).

The following property of the GCW matrix $\boldsymbol{S}$ motivates our procedure for choosing weights.

**Proposition 4.1.** *Assume $l \in \mathbb{Z}_+$ and the setting of permutation synchronization with $G([n],E)$, $E_b$, $\boldsymbol{W}$, $\boldsymbol{X}^*$ and $\tilde{\boldsymbol{X}}$. Assume that $\boldsymbol{W}(i,j) = 0$ for $ij \in E_b$ and $\boldsymbol{W}^l(i,j) > 0$ for $ij \in E$. Then the GCW matrix $\boldsymbol{S}$ satisfies*

$$\boldsymbol{S}^l \oslash (\boldsymbol{W}^l \otimes \mathbf{1}_m) = \boldsymbol{X}^*, \qquad (7)$$

*where the equality is constrained to the blocks $[i,j] \in E$ and $l$ is used for matrix power.*

Our idea is to iteratively estimate the correlation affinity matrix $\boldsymbol{A}^* = \langle \boldsymbol{X}^*, \tilde{\boldsymbol{X}} \rangle_{\text{block}}/m$, by using Proposition 4.1 to approximate $\boldsymbol{X}^*$ with an estimate of $\boldsymbol{S}^l \oslash (\boldsymbol{W}^l \otimes \mathbf{1}_m)$ obtained at each iteration. For simplicity, we assume that $l = 2$. This basic idea is summarized in Algorithm 1 (due to its mentioned equivalence with CEMP [17] we do not propose a new name for it). It initializes the weights by the adjacency matrix. It then performs three steps at each iteration: Estimation of GCW; estimation of the affinity matrix; and estimation of weights.

---

**Algorithm 1** CEMP (reformulated)

---

**Input:** measured relative permutations $\tilde{\boldsymbol{X}}$, Adjacency matrix $\boldsymbol{E}$, total time step $t_0$, increasing $\{\beta_t\}_{t=0}^{t_0}$

$\quad \boldsymbol{W}_{\text{init},(0)} = \boldsymbol{E}$
$\quad$**for** $t = 0 : t_0$ **do**
$\quad\quad \boldsymbol{S}_{\text{init},(t)} = (\boldsymbol{W}_{\text{init},(t)} \otimes \mathbf{1}_m) \odot \tilde{\boldsymbol{X}}$
$\quad\quad \boldsymbol{A}_{\text{init},(t)} = \frac{1}{m} \langle \boldsymbol{S}_{\text{init},(t)}^2 \oslash (\boldsymbol{W}_{\text{init},(t)}^2 \otimes \mathbf{1}_m), \tilde{\boldsymbol{X}} \rangle_{\text{block}}$
$\quad\quad \boldsymbol{W}_{\text{init},(t+1)} = \exp(\beta_t \boldsymbol{A}_{\text{init},(t)})$
$\quad$**end for**
**Output:** $\boldsymbol{A}_{\text{init}} = \boldsymbol{A}_{\text{init},(t)}$

---

We formally explained the second step by using Proposition 4.1 with $l = 2$ and approximating $\boldsymbol{X}^*$ with $\boldsymbol{S}_{\text{init},(t)}^2 \oslash (\boldsymbol{W}_{\text{init},(t)}^2 \otimes \mathbf{1}_m)$. Let us gain some intuition for this formal expression. We claim that (7) encodes the $(l+1)$-cycle consistency relationship. For $l = 2$, i.e., for 3-cycles, this relationship is $\boldsymbol{X}_{ik}^* \boldsymbol{X}_{kj}^* = \boldsymbol{X}_{ij}^*$. In order to explain this claim, we denote $N(ij) = \{k \in [n] : ik, jk \in E\}$ and $N_g(ij) = \{k \in [n] : ik, jk \in E_g\}$. Our approximation, $\boldsymbol{X}_{ij,(t)}^{\text{apprx}}$, for the $(i,j)$th block of $\boldsymbol{X}^*$, $\boldsymbol{X}^*[i,j]$, or equivalently, for $\boldsymbol{S}_{\text{init},(t)}^2 \oslash (\boldsymbol{W}_{\text{init},(t)}^2 \otimes \mathbf{1}_m)[i,j]$ or $\boldsymbol{S}_{\text{init},(t)}^2[i,j]/\boldsymbol{W}_{\text{init},(t)}^2(i,j)$, can be written as

$$\boldsymbol{X}_{ij,(t)}^{\text{apprx}} = \frac{\sum_{k \in N(ij)} \boldsymbol{W}_{\text{init},(t)}(i,k) \boldsymbol{W}_{\text{init},(t)}(k,j) \tilde{\boldsymbol{X}}_{ik} \tilde{\boldsymbol{X}}_{kj}}{\sum_{k \in N(ij)} \boldsymbol{W}_{\text{init},(t)}(i,k) \boldsymbol{W}_{\text{init},(t)}(k,j)} \approx \frac{1}{|N_g(ij)|} \sum_{k \in N_g(ij)} \boldsymbol{X}_{ik}^* \boldsymbol{X}_{kj}^* = \boldsymbol{X}_{ij}^*. \quad (8)$$

Proposition 4.1 provides a condition for making the unexplained approximation in (8) an equality. In view of (8), we estimate $a_{ij}^*$ by $\langle \tilde{\boldsymbol{X}}_{ij}, \boldsymbol{X}_{ij,(t)}^{\text{apprx}} \rangle/m$.

Our third step uses the exponential function. In order to explain this choice, we note that if $\boldsymbol{A}_{\text{init},(t)} \to \boldsymbol{A}^*$, or equivalently $\boldsymbol{X}_{ij,(t)}^{\text{apprx}} \to \boldsymbol{X}_{ij}^*$, as $t \to \infty$, then

$$\boldsymbol{W}_{\text{init},(t+1)}(i,j) \to \exp\left(\beta_t \langle \tilde{\boldsymbol{X}}_{ij}, \boldsymbol{X}_{ij}^* \rangle/m\right) = \exp\left(-\frac{\beta_t}{2m} \|\tilde{\boldsymbol{X}}_{ij} - \boldsymbol{X}_{ij}^*\|_F^2\right) \cdot \exp(\beta_t). \quad (9)$$

Due to the arbitrary normalization of $\boldsymbol{W}_{\text{init},(t+1)}(i,j)$, which is evident from (8), we can ignore the term $\exp(\beta_t)$. We note that this update rule is mathematically equivalent to the heat kernel used in Vector Diffusion Maps [26]. By taking $\beta_t \to \infty$, we obtain that $\exp(-\beta_t \|\boldsymbol{X}_{ij}^* - \tilde{\boldsymbol{X}}_{ij}\|_F^2/2m) \to \mathbf{1}_{\{ij \in E_g\}}$. This will clearly result in equality in (8) (that is, $\boldsymbol{W}_{\text{init},(t)}$ satisfies the requirements of Proposition 4.1) and consequently $\boldsymbol{A}_{\text{init},(t+1)} \to \boldsymbol{A}^*$. Therefore, when $t \to \infty$ and $\beta_t \to \infty$, $\boldsymbol{A}^*$ is a fixed point of Algorithm 1.

Finally, we formulate the mentioned equivalence with CEMP [17] in the more general setting of group synchronization. Consequently, the established theory for CEMP in [17] extends to our procedure (Proposition 4.1 only motivates our procedure but does not justify it). We recall that CEMP directly estimates the corruption levels $\{d(\tilde{\boldsymbol{X}}_{ij}, \boldsymbol{X}_{ij}^*)\}_{ij \in E}$ for some metric $d$. It coincides with our approach when using the metric $d(\boldsymbol{X}, \boldsymbol{Y}) = \|\boldsymbol{X} - \boldsymbol{Y}\|_F^2/(2m)$ for $\boldsymbol{X}, \boldsymbol{Y} \in \mathscr{P}_m$. We note that since $\|\tilde{\boldsymbol{X}}_{ij} - \boldsymbol{X}_{ij}^*\|_F^2 = 2m - 2\langle \tilde{\boldsymbol{X}}_{ij}, \boldsymbol{X}_{ij}^* \rangle$, the corruption level used by CEMP for $ij \in E$ is $1 - a_{ij}^*$, where $a_{ij}^*$ is the affinity of our procedure. Because the details of CEMP for estimating the corruption levels are more involved than our ideas, we prefer not to review them here.

**Proposition 4.2.** *Assume that $\tilde{\boldsymbol{X}}$ represents the measured relative permutations in permutation synchronization, or more generally, the measured relative groups ratios in compact group synchronization, where the group has an orthogonal matrix representation. Assume further the following semimetric on the matrix-represented elements $\boldsymbol{X}$, $\boldsymbol{Y}$: $\|\boldsymbol{X} - \boldsymbol{Y}\|_F^2/(2m)$ (which is metric for the permutation group). Then CEMP with this semimetric and 3-cycles is equivalent to Algorithm 1. If one uses $l$-th powers with $l \geq 2$ in Algorithm 1, then it is equivalent to CEMP with $(l+1)$-cycles and the same semimetric.*

## 4.2 Weighted Least Squares Approximation of Permutations

We assume the approximated weights $\{w_{ij,(t)}\}_{ij\in E}$ at iteration $t\geq 0$, where at $t=0$ the weights are obtained by Algorithm 1. The GCW and normalized GCW matrices are $\boldsymbol{S}_{(t)}$ and $\overline{\boldsymbol{S}}_{(t)}$. Using these weights, one may approximate the absolute permutations as solutions of two different WLS problems. The first one, which we advocate for, aims to solve the following weighted power iterations, which is a weighted analog of (4):

$$\boldsymbol{P}_{i,(t+1)}=\operatorname*{argmin}_{\boldsymbol{P}_i\in\mathscr{P}_m}\left\|\sum_{j\in N(i)}w_{ij,(t)}(\boldsymbol{P}_i-\tilde{\boldsymbol{X}}_{ij}\boldsymbol{P}_{j,(t)})\right\|_F^2\quad\text{for all }i\in[n]. \tag{10}$$

We note that the solution of (10) is $\boldsymbol{P}_{(t+1)}=\operatorname{Proj}(\boldsymbol{S}_{(t)}\boldsymbol{P}_{(t)})$. The second WLS formulation aims to solve

$$\{\boldsymbol{P}_{i,(t+1)}\}_{i\in[n]}=\operatorname*{argmin}_{\{\boldsymbol{P}_i\}_{i\in[n]}\subset\mathscr{P}_m}\sum_{i\in[n]}\left\|\frac{1}{d_{i,(t)}}\sum_{j\in N(i)}w_{ij,(t)}(\boldsymbol{P}_i-\tilde{\boldsymbol{X}}_{ij}\boldsymbol{P}_j)\right\|_F^2, \tag{11}$$

where $d_{i,(t)}$ is the degree of node $i$. To approximately solve (11), one can relax its constraint by requiring that $\boldsymbol{P}^\mathsf{T}\boldsymbol{P}=\boldsymbol{I}_m$ and after projection onto $\mathscr{P}_m$ obtain that

$$\boldsymbol{P}_{(t+1)}=\operatorname{Proj}\left(\operatorname*{argmin}_{\boldsymbol{P}^\mathsf{T}\boldsymbol{P}=\boldsymbol{I}_m}\left\|\boldsymbol{P}-\overline{\boldsymbol{S}}_{(t)}\boldsymbol{P}\right\|_F^2\right). \tag{12}$$

The columns of the solution of the minimization in (12) are exactly the top $m$ eigenvectors of $\overline{\boldsymbol{S}}_{(t)}$. An analogue of (11) for $SE(3)$ synchronization appears in [2]. We recommend using (10) over (11) as it is faster and often more accurate in practice. However, since (12) does not require a prior estimate for $\boldsymbol{P}$, we use it for the initial estimate of the block matrix of absolute permutations, $\boldsymbol{P}_{(1)}$. We report results for both methods.

## 4.3 Iteratively Reweighted Graph Connection Laplacian

We combine together ideas of §4.1 and §4.2 to formulate the Iteratively Reweighted Graph Connected Laplacian (IRGCL) procedure in Algorithm 2. The initial affinity matrix is computed by CEMP and the initial

---

**Algorithm 2** Iteratively Reweighted Graph Connection Laplacian (IRGCL)

---
**Input:** $\tilde{\boldsymbol{X}}$, $\{\beta_t\}_{t=0}^{t_0}\nearrow$, $\{\alpha_t\}_{t=1}^{t_{\max}}\nearrow$, $\{\lambda_t\}_{t=1}^{t_{\max}}\nearrow$, $F:\mathbb{R}^{n\times n}\to\mathbb{R}^{n\times n}$ (default: $F(\boldsymbol{A})=\boldsymbol{A}$)
    $\boldsymbol{A}_{(0)}=\operatorname{CEMP}(\tilde{\boldsymbol{X}},\{\beta_t\}_{t=0}^{t_0})$
    $\boldsymbol{W}_{(0)}=F(\boldsymbol{A}_{(0)})$
    $\boldsymbol{P}_{(1)}=\operatorname{WLS}(\tilde{\boldsymbol{X}},\boldsymbol{W}_{(0)})$ by solving (12)
    **for** $t=1:t_{\max}$ **do**
        $\boldsymbol{X}_{(t)}=\boldsymbol{P}_{(t)}\boldsymbol{P}_{(t)}^\mathsf{T}$
        $\boldsymbol{A}_{1,(t)}=\frac{1}{m}\langle\boldsymbol{X}_{(t)},\tilde{\boldsymbol{X}}\rangle_{\text{block}}$
        $\boldsymbol{W}_{1,(t)}=\exp(\alpha_t\boldsymbol{A}_{1,(t)})$
        $\boldsymbol{S}_{(t)}=(\boldsymbol{W}_{1,(t)}\otimes\boldsymbol{1}_m)\odot\tilde{\boldsymbol{X}}$
        $\boldsymbol{A}_{2,(t)}=\frac{1}{m}\langle\boldsymbol{S}_{(t)}^2\oslash(\boldsymbol{W}_{1,(t)}^2\otimes\boldsymbol{1}_m),\tilde{\boldsymbol{X}}\rangle_{\text{block}}$
        $\boldsymbol{A}_{(t)}=(1-\lambda_t)\boldsymbol{A}_{1,(t)}+\lambda_t\boldsymbol{A}_{2,(t)}$
        $\boldsymbol{W}_{(t)}=F(\boldsymbol{A}_{(t)})$
        $\boldsymbol{P}_{(t+1)}=\operatorname{WLS}(\tilde{\boldsymbol{X}},\boldsymbol{W}_{(t)})$ by solving (10) (or possibly (12))
    **end for**
**Output:** estimated absolute permutations $\boldsymbol{P}_{(t+1)}$

---

block of absolute permutations $\boldsymbol{P}_{(1)}$ is obtained by solving (12) (we explained above why (10) cannot be used for initialization). At next iterations, the affinity matrix is obtained as a convex combination of two affinity matrices as follows: $(1-\lambda_t)\boldsymbol{A}_{1,(t)}+\lambda_t\boldsymbol{A}_{2,(t)}$. The matrix $\boldsymbol{A}_{1,(t)}$ is directly obtained by the newly estimated absolute permutations. Its use is similar to that of standard IRLS. Indeed, in IRLS the residuals are updated, but here we work with dot products instead of residuals (squared norms). It is easy to note that the matrix $\boldsymbol{A}_{2,(t)}$ is updated in a similar way to the procedure described in Algorithm 1. The permutations in $\boldsymbol{P}_{(t+1)}$ are updated by a WLS procedure. When using (12) for this purpose, we name the algorithm IRGCL-S (S for spectral). When using (10) instead, we name the algorithm IRGCL-P (P for power iterations; this is our recommended choice).

The main difference between IRLS and our IRGCL is that IRGCL uses both the 1st and 2nd order edge affinities: $\boldsymbol{A}_{1,(t)}$ and $\boldsymbol{A}_{2,(t)}$, defined in the algorithm, to approximate $\boldsymbol{A}^*$ and computes the WLS

weights $\boldsymbol{W}_{(t)}$. However, standard IRLS computes the WLS weights from only the 1st order affinities (or equivalently residuals; see (5)), which are unreliable under high corruption. Indeed, we can write $\boldsymbol{A}_{1,(t)}(i,j) = 1 - \|\boldsymbol{X}_{ij,(t)} - \tilde{\boldsymbol{X}}_{ij}\|_F^2 / 2m$ and note that the approximation of $\boldsymbol{A}^*$ by $\boldsymbol{A}_{1,(t)}$ can be poor when $\boldsymbol{X}_{(t)}$ deviates from $\boldsymbol{X}^*$. Therefore, we address this issue by gradually incorporating the 2nd order affinities (in $\boldsymbol{A}_{2,(t)}$), which encode the 3-cycle consistency information (see explanation before (8)). We do this by increasing $\lambda_t$ towards 1 as $t$ increases. However, incorporating the information from $\boldsymbol{A}_{1,(t)}$ during the first few iterations can accelerate the convergence; we thus start with $\lambda_1 = 0.5$.

Few more technical comments on Algorithm 2 are as follows. It contains two types of edge weights: $\boldsymbol{W}_{1,(t)}$ for the CEMP-like reweighting for estimating $\boldsymbol{A}_{2,(t)}$ and $\boldsymbol{W}_{(t)}$ for the WLS formulation that is used to solve the absolute permutations. The latter weights are estimated using the affinity $(1-\lambda_t)\boldsymbol{A}_{1,(t)} + \lambda_t\boldsymbol{A}_{2,(t)}$. For simplicity and to avoid additional parameters, we assume that $F(\boldsymbol{A}_{(t)}) = \boldsymbol{A}_{(t)}$. For the same reason, we only incorporate second order affinities and avoid higher order ones. The default parameters are described in Section 6.

We further illustrate IRGCL in Figure 2 in the supplementary material. As is evident from this Figure, its basic idea is similar to the MPLS [24] algorithm that was pursued for the different problem of rotation synchronization. Nevertheless, there are two main differences between the two implementations. First, the reweighting function $F$ of [24] is sensitive to zero residuals and requires iterative truncation that introduces additional parameters. Second, [24] enforces $\lambda_t \to 0$ (as opposed to $\lambda_t \to 1$) and thus emphasizes the standard IRLS procedure, except for the first few iterations. The latter choice is mainly due to numerical experience with real data. We further support this choice when explaining below the need for cycle information in PS due to zero residuals.

We do not have guarantees for Algorithm 2, but we believe it is successful due to the following properties. First, it utilizes information from 3-cycles (reflected in the powers of the GCW matrix). We believe that this decreases the sensitivity to its initialization (this is evident in numerical experiments). We further remark that the resulted estimate is more robust to corruption with nonuniform graph topology. Indeed, using the cycle information allows messages from $G([n], E_g)$ to propagate through the entire graph more easily and consequently correct severely corrupted subgraphs with nonuniform topology. This claim is supported by the experiments with nonuniform corruption. Second, since the 3-cycle consistency information helps more faithfully recover the underlying corruption, it provides more accurate weights and consequently a better approximation to the relaxation of the discrete WLS problem. At last, the elements of $\boldsymbol{A}_{2,(t)}$ for each $ij \in E$ are essentially weighted averages (ideally, expectations) of the 3-cycle consistency. The corresponding expectations are continuous and thus our reweighting scheme smooths the space of edge weights. This may make the algorithm less likely to get stuck.

## 5 Theoretical Guarantees for Nonuniform Corruption

As we mentioned in Section 1, previous work mainly addressed uniform corruption models, where the degrees of the corrupted graph, $G([n], E_b)$ have little variation and the corruption probabilities are uniform (see e.g., [8]). The theory of [17] considers arbitrarily corrupted relative permutations, however, it restricts the maximal ratio of corrupted cycles and consequently restricts the degree of nodes in $E_b$. Here we consider a toy model with non-uniform graph topology (with large variations in the degrees of the subgraph $G([n], E_b)$) and with a relatively general class of distributions for the absolute and corrupted relative permutations.

We refer to our model as the superspreader corruption model. In this model, the bad edges are connected to a single node $i_0$, so the set $E_b$ has a "star-shaped" topology. Moreover, we assume that most of the neighboring edges of $i_0$ are corrupted. The distributions for $\{\boldsymbol{P}_i^*\}_{i\in[n]}$ and $\{\tilde{\boldsymbol{X}}_{ij}\}_{ij\in E_b}$ are general. We show that under this model and an additional mild generic condition on the latter distribution, least-squares type methods, including PPM, may fail, whereas CEMP (Algorithm 1) is able to achieve accurate estimation of $\boldsymbol{A}^*$ in one iteration as long as $n$ and its parameter $\beta_0$ are sufficiently large. We remark that the generic theory established for CEMP in [17] does not apply to the superspreader model. Our ideas of proof are also different from those of [17].

We first formulate this model. We then formulate the theorem, which is proved in the supplementary material.

**Definition 5.1.** *The superspreader corruption model with parameters $n \in \mathbb{N}$, $m \in \mathbb{N}$ and $0 < \varepsilon, p \leq 1 \in \mathbb{R}$; distributions $\mathcal{D}_P$ and $\mathcal{D}_X$ on $\mathscr{P}_m$; and superspreader node $i_0 \in [n]$ is a probabilistic model with the following components: an Erdős-Rényi graph $G(n,p)$ where $p$ is probability of connection; ground-truth absolute permutations $\{\boldsymbol{P}_i^*\}_{i\in[n]}$ i.i.d. sampled from $\mathcal{D}_P$; a set $E_b$, whose edges are of the form $i_0 j$, where $j$ is randomly sampled from $N(i_0)$ with probability $1-\varepsilon$; and corrupted measurements of relative permutations $\{\tilde{\boldsymbol{X}}_{ij}\}_{ij\in E_b}$, such that $\tilde{\boldsymbol{X}}_{ij}$ is i.i.d sampled from $\mathcal{D}_X$ and for $ij \in E_g$, $\tilde{\boldsymbol{X}}_{ij} = \boldsymbol{X}_{ij}^* \equiv \boldsymbol{P}_i^* \boldsymbol{P}_j^{*\intercal}$.*

**Theorem 5.2.** *Assume data generated by the superspreader corruption model with node $i_0$, parameters $n$, $m$ and $0 < \varepsilon, p \leq 1$, distributions $\mathcal{D}_P$ and $\mathcal{D}_X$, and ground-truth and measured relative permutations $\{\boldsymbol{X}_{ij}^*\}_{ij\in E}$ and $\{\tilde{\boldsymbol{X}}_{ij}\}_{ij\in E}$, respectively. Let $\boldsymbol{A}^* = \langle \boldsymbol{X}^*, \tilde{\boldsymbol{X}} \rangle_{block} / m$, $\mu = \mathbb{E}\left(\|\tilde{\boldsymbol{X}}_{i_0 j} - \boldsymbol{X}_{i_0 j}^*\|_F^2 \mid j \in N_b(i_0)\right) / (2m)$, and*

assume that for all $k \in N_b(i_0)$

$$\mathbb{E}\Big(\|\tilde{\boldsymbol{X}}_{i_0 j} - \boldsymbol{X}^*_{i_0 j}\|^2_F \,|\, j \in N_b(i_0)\Big) \leq \mathbb{E}\Big(\|\tilde{\boldsymbol{X}}_{ki_0}\tilde{\boldsymbol{X}}_{i_0 j} - \boldsymbol{X}^*_{kj}\|^2_F \,|\, j,k \in N_b(i_0)\Big). \tag{13}$$

*Then, for $n = \Omega(1/(\mu^2 \varepsilon^2 p^2))$, and $\beta_0$ of CEMP, $\boldsymbol{A}_{init,(1)}$ obtained by CEMP with one iteration satisfies w.h.p.*

$$\|\boldsymbol{A}_{init,(1)} - \boldsymbol{A}^*\|_\infty \leq (2-\epsilon)\Big(2-\epsilon+\epsilon e^{\beta_0 \mu \epsilon/2}\Big)^{-1}. \tag{14}$$

*On the other hand, for sufficiently small $\varepsilon$, large $n$ and some choices for $\mathcal{D}_X$, any least squares method, in particular PPM, does not result in good approximation of $\{\boldsymbol{X}^*_{ij}\}_{ij \in E}$ and subsequent good estimation of $\boldsymbol{A}^*$.*

The proof easily clarifies that condition (13) means that when the number of corrupted edges in a 3-cycle is enlarged from 1 to 2, then the cycle consistency decreases on average. A more precise statement of the second part of the theorem is that if $\|\mathbb{E}(\tilde{\boldsymbol{X}}_{i_0 j}\boldsymbol{P}^*_j) - \boldsymbol{P}_{\text{crpt}}\|_F < \varepsilon_0/2$ for $\varepsilon_0 > 0$, such that $2\varepsilon\sqrt{2m} + (1-2\varepsilon)\varepsilon_0 < 1$, and $\boldsymbol{P}_{\text{crpt}} \neq \boldsymbol{P}^*_{i_0}$, then PPM (and similarly any least squares method) cannot recover the ground-truth permutations w.h.p. for $n$ sufficiently large.

# 6   Numerical Experiments

Using synthetic and real data, we compared IRGCL-S&P (IRGCL-S and IRGCL-P) with the following methods for PS: Spectral [22]; PPM [7]; IRLS-Cauchy-S&P (two methods that adapt the idea of [2] to PS, while solving the WLS problem by either (10) for IRLS-Cauchy-P or (12) for IRLS-Cauchy-S); MatchLift [8] and MatchALS [34]. For the last two methods we used the codes from `https://github.com/zju-3dv/multiway` and their default choices. We implemented the rest of the methods using the default choices in the corresponding papers. We use the following parameters for IRGCL-S&P: $t_0 = 5$, $t_{\max} = 100$, $\beta_t = \min(2^t, 40)$, $\alpha_t = \min(1.2^{t-1}, 40)$, $\lambda_t = t/(t+1)$ and $F(\boldsymbol{A}) = \boldsymbol{A}$. We stop the algorithm whenever $\boldsymbol{P}_{(t+1)} = \boldsymbol{P}_{(t)}$.

In §6.1 we report results on synthetic data with a nonuniform corruption model, where the supplementary material further includes results with uniform corruption. In §6.2 we include results for real data.

## 6.1   Nonuniform Corruption Models

The following two models involve nonuniform corruption. For both models, we choose $n = 100$, $m = 10$ and assume an underlying complete graph $G([n], E)$. Experiments with a more general Erdős-Rényi graph are reported in the supplementary material. We independently sample $n_c$ nodes and for each sampled node we independently corrupt its $m_c$ incident edges. We remark that $n_c = 1$ corresponds to our superspreader corruption model. We let $\{\boldsymbol{P}^c_i\}_{i \in [n]}$ be i.i.d. sampled from the Haar measure on $\mathscr{P}_m$, $\text{Haar}(\mathscr{P}_m)$. We next describe the generation of $\tilde{\boldsymbol{X}}_{ij}$, where $ij \in E_b$, in the two models. It is maliciously designed so that the distribution of $\tilde{\boldsymbol{X}}_{ij}\boldsymbol{P}^*_j$ is no longer concentrated around $\boldsymbol{P}^*_i$, but biased towards some other permutation matrix.
*1. Local Biased Corruption Model (LBC):* For each $ij \in E_b$,

$$\tilde{\boldsymbol{X}}_{ij} = \begin{cases} \boldsymbol{P}^c_i \boldsymbol{P}^{c\mathsf{T}}_j, & \text{if } \langle \boldsymbol{P}^c_i \boldsymbol{P}^{c\mathsf{T}}_j, \boldsymbol{P}^*_i \boldsymbol{P}^{*\mathsf{T}}_j \rangle \leq 1; \\ \boldsymbol{X}_{ij} \sim \text{Haar}(\mathscr{P}_m), & \text{otherwise.} \end{cases} \tag{15}$$

Note that $\boldsymbol{P}^c_i \boldsymbol{P}^{c\mathsf{T}}_j$ are self-consistent and since $\langle \boldsymbol{P}^c_i \boldsymbol{P}^{c\mathsf{T}}_j, \boldsymbol{P}^*_i \boldsymbol{P}^{*\mathsf{T}}_j \rangle \leq 1$, they tend to be far away from the ground-truth $\boldsymbol{P}^*_i \boldsymbol{P}^{*\mathsf{T}}_j$, and therefore the overall distribution of $\tilde{\boldsymbol{X}}_{ij}\boldsymbol{P}^*_j$ is far away from $\boldsymbol{P}^*_i$.
*2. Local Adversarial Corruption Model (LAC):* For each $ij \in E_b$: $\tilde{\boldsymbol{X}}_{ij} = \boldsymbol{Q}^c_{ij}\boldsymbol{P}^{*\mathsf{T}}_j$, where $\boldsymbol{Q}^c_{ij}$ is sampled by randomly permuting 3 columns of the $m \times m$ identity matrix. We remark that the LAC model is even more malicious, since $\tilde{\boldsymbol{X}}_{ij}\boldsymbol{P}^*_j$ explicitly concentrates around the identity matrix.

We fix $m_c = 90$ for LBC and $m_c = 60$ for LAC. We use the error $\sum_{ij \in E_b}\|\hat{\boldsymbol{X}}_{ij} - \boldsymbol{X}^*_{ij}\|^2_F \,/\, \sum_{ij \in E_b}\|\boldsymbol{X}^*_{ij}\|^2_F$ to compare the different methods. We created 20 random samples from each model and we computed average errors and standard deviations for $n_c = 1,...,6$. Figure 1 reports these average errors for the two different models, while designating standard deviations by error bars.

We note that both methods are able to achieve near exact recovery under all tested values of $n_c$. In particular, they can exactly recover the ground-truth permutations under the super malicious LAC model, and outperform all other methods. We remark that both IRLS-Cauchy-S&P perform better than Spectral and PPM. However, their improvement is limited and cannot achieve exact recovery. MatchLift and MatchALS are better than other least squares methods. However, they require hundreds of iterations and are thus slow.

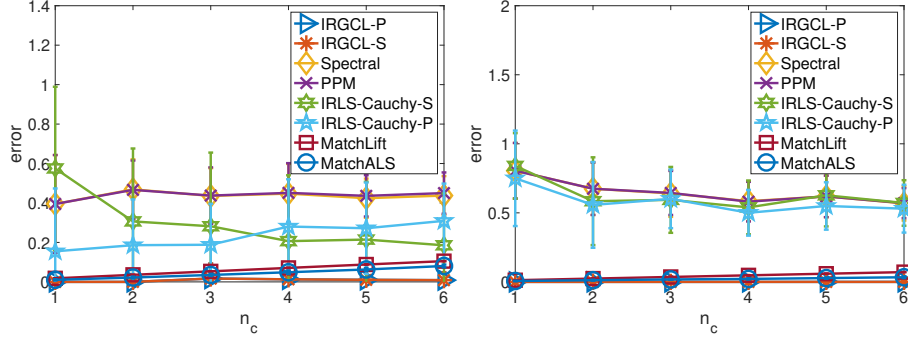

Figure 1: Average matching error under the LBC model (left) and LAC model (right).

## 6.2 Real Dataset

We compare the performance of the different methods on the Willow database [9], which consists of 5 image datasets. Each image dataset contains 40-108 images of the same object. We use the same method suggested by [30] to extract CNN features from 10 annotated keypoints for each image through AlexNet [16]. The candidate for the initial matching is obtained by applying the Hungarian algorithm on the feature similarity matrix, following the same procedure as [30]. However, the obtained initial matching is ill-posed for permutation synchronization. For example, given the initial matching obtained using the car dataset, there are 8 out of 40 nodes whose all neighboring edges are severely corrupted. That is, there is no chance to recover correct information of those nodes. To make those datasets well-posed to PS solvers, we only use the relative permutation $\tilde{X}_{ij}$ between the nodes $i$ and $j$ whose incident edges are not completely corrupted. IRLS-Cauchy-P&S were comparable and we thus only report IRLS-Cauchy-S, while referring to it as IRLS. We also report the estimation error for $P_{(1)}$ in Algorithm 2 (we call it IRGCL-init and further test it with artificial data in the supplementary material). We do not compare with [30] since it requires additional geometric information from the pixel coordinates of keypoints. We report the relative estimation error $\sum_{i \neq j} \|\hat{X}_{ij} - X^*_{ij}\|^2_F / \sum_{i \neq j} \|X^*_{ij}\|^2_F$ of different methods in Table 1. We note that the four data sets which exclude FACE are highly corrupted (in view of their "Input" parameter).

| Datasets | $n$ | Input | Spectral [22] | MLift [8] | MALS [34] | PPM [14] | IRLS [2] | IRGCL-init ours | IRGCL-S ours | IRGCL-P ours |
|---|---|---|---|---|---|---|---|---|---|---|
| Car | 32 | 0.41 | 0.23 | 0.17 | 0.14 | 0.11 | 0.16 | 0.14 | 0.14 | **0.091** |
| Duck | 30 | 0.46 | 0.20 | 0.26 | 0.22 | 0.20 | **0.19** | **0.19** | **0.19** | 0.21 |
| Face | 108 | 0.14 | 0.042 | 0.071 | 0.057 | 0.049 | 0.042 | **0.039** | **0.039** | 0.051 |
| Motorbike | 14 | 0.55 | 0.46 | 0.49 | 0.48 | 0.44 | 0.46 | 0.41 | 0.42 | **0.33** |
| Winebottle | 56 | 0.43 | 0.27 | 0.24 | 0.22 | 0.24 | 0.24 | 0.22 | 0.22 | **0.21** |

Table 1: Matching performance comparison using the Willow datasets.

Our methods IRGCL-S, IRGCL-init and IRGCL-P are still able to achieve reasonable improvement over Spectral and PPM respectively. Among the least squares methods, Spectral and MatchLift perform the worst on average, and PPM performs the best. We remark that IRLS with Cauchy weights does not have a significant advantage over the least squares methods. We note that IRGCL-S and IRGCL-init perform similarly. Furthermore, on average IRGCL-P performs the best, especially for the highly corrupted datasets (excluding FACE).

## 7 Conclusion

We proposed an iterative method for robustly solving multi-object matching. It overcomes the limitations of both IRLS and common least squares methods under nonuniform corruption models. We demonstrated through both experiments and theory the advantage of directly exploiting cycle-consistency information to guide the convergence of our non-convex optimization algorithm. There are several interesting future directions. First of all, although our work focuses on permutation synchronization, its ideas can be generalized to the setting of partial matching, which has more applications in structure from motion. Second, we believe that one can borrow ideas from the theory of graph connection Laplacian and vector diffusion maps in order to establish exact recovery guarantees for our method under different corruption models.

# 8 Broader Impact

Our proposed algorithms and ideas can be integrated in common 3D reconstruction software. Three-dimensional reconstruction has important applications in autonomous driving, virtual reality and augmented reality. In the past decade, the 3D reconstruction community has been switching from incremental reconstruction procedures to global optimization schemes [21]. We thus globally estimate correlations to provide consistent image matches as initial data for common global reconstruction pipelines. In order to address real applied scenarios of high corruption, it is important to further develop and utilize robust estimation methods within real-time 3D reconstruction. In addition to developing robust methods, we also provide some theoretical guarantees for a special setting of nonuniform corruption. Another important reason for detecting abnormal data in an unsupervised and interpretable way is to alleviate the vulnerability of deep learning based methods to adversarial attacks. Our work takes a step towards this aim through robust extraction of image or camera correspondence information without pre-training. This work is of interest to a broad community of machine learners that care about and use robustness, discrete optimization methods and iteratively reweighted least squares (IRLS). In fact, we show that the common IRLS method does not work well in our setting and explain how to carefully modify it. We use core and well-established testing methods and prove various mathematical propositions.

## Acknowledgement

This work was supported by NSF award DMS-18-21266.

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
