[Supplementary Material]

# Supplementary Material

In §A we provide proofs of Propositions 4.2 and 4.1 and Theorem 5.2. In §B we provide an illustration that explains IRGCL and results of additional experiments on synthetic data generated by both uniform and nonuniform corruption models. We follow the equation and figure numbers of the main manuscript. We thus start with equation number (16) and Figure 2.

## A    Proofs

In this section we provide the proofs to all theoretical statements in the main manuscript. We find it more convenient to first establish Proposition 4.1 and then use the simple idea of the proof when establishing Proposition 4.2. At last, we prove Theorem 5.2.

### A.1    Proof of Proposition 4.1

We first note that $(\boldsymbol{S}^l \oslash (\boldsymbol{W}^l \otimes \boldsymbol{1}_m))[i,j] = \frac{1}{\boldsymbol{W}^l(i,j)} \boldsymbol{S}^l[i,j]$. where

$$\boldsymbol{W}^l(i,j) = \sum_{k_1 \in [n]} \sum_{k_2 \in [n]} \cdots \sum_{k_{l-1} \in [n]} w_{ik_1} w_{k_1 k_2} \cdots w_{k_{l-1}j} \tag{16}$$

$$\boldsymbol{S}^l[i,j] = \sum_{k_1 \in [n]} \sum_{k_2 \in [n]} \cdots \sum_{k_{l-1} \in [n]} w_{ik_1} w_{k_1 k_2} \cdots w_{k_{l-1}j} \tilde{\boldsymbol{X}}_{ik_1} \tilde{\boldsymbol{X}}_{k_1 k_2} \cdots \tilde{\boldsymbol{X}}_{k_{l-1}j}. \tag{17}$$

We note that for any pair of nodes $i,j \in [n]$, $w_{ij} \tilde{\boldsymbol{X}}_{ij} = w_{ij} \boldsymbol{X}_{ij}^*$. Indeed, if $ij \notin E$, then $w_{ij} = 0$; if $ij \in E_g$, then $\tilde{\boldsymbol{X}}_{ij} = \boldsymbol{X}_{ij}^*$; and if $ij \in E_b$, then $\boldsymbol{W}(i,j) = 0$ by the assumption of the proposition. Combining this observation with (16) and (17) and then applying the cycle-consistency of $\{\boldsymbol{X}_{ij}^*\}_{ij \in E}$, we obtain that for $ij \in E$,

$$(\boldsymbol{S}^l \oslash (\boldsymbol{W}^l \otimes \boldsymbol{1}_m))[i,j] = \frac{\sum_{k_1 \in [n]} \cdots \sum_{k_{l-1} \in [n]} w_{ik_1} \cdots w_{k_{l-1}j} \boldsymbol{X}_{ik_1}^* \boldsymbol{X}_{k_1 k_2}^* \cdots \boldsymbol{X}_{k_{l-1}j}^*}{\sum_{k_1 \in [n]} \cdots \sum_{k_{l-1} \in [n]} w_{ik_1} \cdots w_{k_{l-1}j}}$$

$$= \frac{\sum_{k_1 \in [n]} \cdots \sum_{k_{l-1} \in [n]} w_{ik_1} \cdots w_{k_{l-1}j} \boldsymbol{X}_{ij}^*}{\sum_{k_1 \in [n]} \cdots \sum_{k_{l-1} \in [n]} w_{ik_1} \cdots w_{k_{l-1}j}} = \boldsymbol{X}_{ij}^*.$$

### A.2    Proof of Proposition 4.2

We first introduce the following definitions and notation for describing the original version of CEMP [17]. Let $N_{ij}$ be the set of all $(l+1)$-cycles that contain $ij$. Any cycle in $N_{ij}$ can be represented as $L := \{ik_1, k_1 k_2, \ldots, k_{l-1}j, ji\}$. Using the metric stated in the proposition, the cycle inconsistency (proposed in [17]) for each $L \in N_{ij}$ is defined as

$$d_L := \|\tilde{\boldsymbol{X}}_{ik_1} \tilde{\boldsymbol{X}}_{k_1 k_2} \tilde{\boldsymbol{X}}_{k_2 k_3} \ldots \tilde{\boldsymbol{X}}_{k_{l-1}j} \tilde{\boldsymbol{X}}_{ji} - \boldsymbol{I}_m\|_F^2 / 2m. \tag{18}$$

The original version of CEMP with $(l+1)$-cycles is iterated over $t \geq 0$ using the following message passing procedure (see (10) and (35) of [17]):

$$s_{ij,(0)} = \sum_{L \in N_{ij}} d_L / |N_{ij}| \tag{19}$$

and

$$s_{ij,(t+1)} = \frac{\sum_{L \in N_{ij}} \prod_{ab \in L \setminus \{ij\}} \exp(-\beta_t s_{ab,(t)}) d_L}{\sum_{L \in N_{ij}} \prod_{ab \in L \setminus \{ij\}} \exp(-\beta_t s_{ab,(t)})} \quad \text{for } ij \in E. \tag{20}$$

We use here a generalized version of Algorithm 1 with power $l \geq 2$. That is, we replace the step $\boldsymbol{A}_{\text{init},(t)} = \frac{1}{m} \langle \boldsymbol{S}_{\text{init},(t)}^2 \oslash (\boldsymbol{W}_{\text{init},(t)}^2 \otimes \boldsymbol{1}_m), \tilde{\boldsymbol{X}} \rangle_{\text{block}}$ in Algorithm 1 with

$$\boldsymbol{A}_{\text{init},(t)} = \frac{1}{m} \langle \boldsymbol{S}_{\text{init},(t)}^l \oslash (\boldsymbol{W}_{\text{init},(t)}^l \otimes \boldsymbol{1}_m), \tilde{\boldsymbol{X}} \rangle_{\text{block}} \quad \text{for } l \geq 2. \tag{21}$$

To prove the equivalence between Algorithm 1 with $l \geq 2$ and CEMP with the chosen metric, we show that at each iteration $t \geq 0$, $\boldsymbol{A}_{\text{init},(t)}(i,j) = 1 - s_{ij,(t)}$, where $\boldsymbol{A}_{\text{init},(t)}(i,j)$ is obtained by Algorithm 1 and $s_{ij,(t)}$ by the original CEMP. We verify this by induction. For simplicity, we denote $\boldsymbol{W}_{\text{init},(t)}(i,j)$ by $q_{ij,(t)}$. For $t = 0$, we

first apply (21), we then combine (16) and (17) and at last apply basic algebraic manipulations to obtain that

$$\boldsymbol{A}_{\text{init},(0)}(i,j) = \frac{1}{m}\left\langle \left(\boldsymbol{S}^l_{\text{init},(0)} \oslash (\boldsymbol{W}^l_{\text{init},(0)} \otimes \boldsymbol{1}_m)\right)[i,j], \tilde{\boldsymbol{X}}_{ij}\right\rangle$$

$$= \frac{1}{m}\left\langle \frac{\sum_{k_1\in[n]}\cdots\sum_{k_{l-1}\in[n]}q_{ik_1,(0)}\cdots q_{k_{l-1}j,(0)}\tilde{\boldsymbol{X}}_{ik_1}\tilde{\boldsymbol{X}}_{k_1k_2}\cdots\tilde{\boldsymbol{X}}_{k_{l-1}j}}{\sum_{k_1\in[n]}\cdots\sum_{k_{l-1}\in[n]}q_{ik_1,(0)}\cdots q_{k_{l-1}j,(0)}}, \tilde{\boldsymbol{X}}_{ij}\right\rangle$$

$$= \frac{1}{m}\frac{\sum_{k_1\in[n]}\cdots\sum_{k_{l-1}\in[n]}q_{ik_1,(0)}\cdots q_{k_{l-1}j,(0)}\left\langle\tilde{\boldsymbol{X}}_{ik_1}\tilde{\boldsymbol{X}}_{k_1k_2}\cdots\tilde{\boldsymbol{X}}_{k_{l-1}j}, \tilde{\boldsymbol{X}}_{ij}\right\rangle}{\sum_{k_1\in[n]}\cdots\sum_{k_{l-1}\in[n]}q_{ik_1,(0)}\cdots q_{k_{l-1}j,(0)}}$$

$$= 1 - \frac{1}{m}\frac{\sum_{k_1\in[n]}\cdots\sum_{k_{l-1}\in[n]}q_{ik_1,(0)}\cdots q_{k_{l-1}j,(0)}\left(m - \left\langle\tilde{\boldsymbol{X}}_{ik_1}\cdots\tilde{\boldsymbol{X}}_{k_{l-1}j}, \tilde{\boldsymbol{X}}_{ij}\right\rangle\right)}{\sum_{k_1\in[n]}\cdots\sum_{k_{l-1}\in[n]}q_{ik_1,(0)}\cdots q_{k_{l-1}j,(0)}}.$$

We further simplify this equation as follows. We first use the fact that

$$m - \langle\boldsymbol{X},\boldsymbol{Y}\rangle = \|\boldsymbol{X}-\boldsymbol{Y}\|_F^2/2 \;\text{ for }\; \boldsymbol{X},\boldsymbol{Y}\in\mathscr{O}_m. \tag{22}$$

We then use the facts that for $\boldsymbol{X}, \boldsymbol{Y}, \boldsymbol{Z}\in\mathscr{O}_m$, $\|\boldsymbol{X}-\boldsymbol{Y}\|_F = \|\boldsymbol{X}\boldsymbol{Z}-\boldsymbol{Y}\boldsymbol{Z}\|_F$ and $\tilde{\boldsymbol{X}}_{ij}\tilde{\boldsymbol{X}}_{ji}=\boldsymbol{I}_m$. Next, we apply (18) and the fact that due to the initialization of $\boldsymbol{W}_{\text{init},(0)}$ by Algorithm 1, $q_{ij,(0)}=1$ for $ij\in E$. At last, we apply (19) and result in the desired relationship when $t=0$:

$$\boldsymbol{A}_{\text{init},(0)}(i,j)$$

$$= 1 - \frac{1}{2m}\frac{\sum_{k_1\in[n]}\cdots\sum_{k_{l-1}\in[n]}q_{ik_1,(0)}\cdots q_{k_{l-1}j,(0)}\left\|\tilde{\boldsymbol{X}}_{ik_1}\tilde{\boldsymbol{X}}_{k_1k_2}\cdots\tilde{\boldsymbol{X}}_{k_{l-1}j}-\tilde{\boldsymbol{X}}_{ij}\right\|_F^2}{\sum_{k_1\in[n]}\cdots\sum_{k_{l-1}\in[n]}q_{ik_1,(0)}\cdots q_{k_{l-1}j,(0)}}$$

$$= 1 - \frac{1}{2m}\frac{\sum_{k_1\in[n]}\cdots\sum_{k_{l-1}\in[n]}q_{ik_1,(0)}\cdots q_{k_{l-1}j,(0)}\left\|\tilde{\boldsymbol{X}}_{ik_1}\tilde{\boldsymbol{X}}_{k_1k_2}\cdots\tilde{\boldsymbol{X}}_{k_{l-1}j}\tilde{\boldsymbol{X}}_{ji}-\boldsymbol{I}_m\right\|_F^2}{\sum_{k_1\in[n]}\cdots\sum_{k_{l-1}\in[n]}q_{ik_1,(0)}\cdots q_{k_{l-1}j,(0)}}$$

$$= 1 - \frac{\sum_{L\in N_{ij}}d_L}{|N_{ij}|} = 1 - s_{ij,(0)}. \tag{23}$$

Next, assuming that for all $ij \in E$ $\boldsymbol{A}_{\text{init},(t)}(i, j) = 1 - s_{ij,(t)}$, we show that all $ij \in E$ $\boldsymbol{A}_{\text{init},(t+1)}(i, j) = 1 - s_{ij,(t+1)}$. We first derive an identity for $\boldsymbol{A}_{\text{init},(t+1)}(i, j)$ which is similar to the second equality of (23) by following the same arguments. We then apply (18) with compact notation for the multiplication of the different weights. Next, we use the weights assigned by Algorithm 1 for $t+1$, that is, $\boldsymbol{W}_{\text{init},(t+1)} = \exp(\beta_t\boldsymbol{A}_{\text{init},(t)})$. Next, we use the induction assumption $\boldsymbol{A}_{\text{init},(t)}(i,j)=1-s_{ij,(t)}$. We then apply basic algebraic manipulations and, at last, use (20) to conclude the induction argument as follows:

$$\boldsymbol{A}_{\text{init},(t+1)}(i,j)$$

$$= 1 - \frac{1}{2m}\frac{\sum_{k_1\in[n]}\cdots\sum_{k_{l-1}\in[n]}q_{ik_1,(t+1)}\cdots q_{k_{l-1}j,(t+1)}\left\|\tilde{\boldsymbol{X}}_{ik_1}\tilde{\boldsymbol{X}}_{k_1k_2}\cdots\tilde{\boldsymbol{X}}_{k_{l-1}j}\tilde{\boldsymbol{X}}_{ji}-\boldsymbol{I}_m\right\|_F^2}{\sum_{k_1\in[n]}\cdots\sum_{k_{l-1}\in[n]}q_{ik_1,(t+1)}\cdots q_{k_{l-1}j,(t+1)}}$$

$$= 1 - \frac{\sum_{L\in N_{ij}}\prod_{ab\in L\setminus\{ij\}}q_{ab,(t+1)}d_L}{\sum_{L\in N_{ij}}\prod_{ab\in L\setminus\{ij\}}q_{ab,(t+1)}}$$

$$= 1 - \frac{\sum_{L\in N_{ij}}\prod_{ab\in L\setminus\{ij\}}\exp\big(\beta_t\boldsymbol{A}_{\text{init},(t)}(a,b)\big)d_L}{\sum_{L\in N_{ij}}\prod_{ab\in L\setminus\{ij\}}\exp\big(\beta_t\boldsymbol{A}_{\text{init},(t)}(a,b)\big)}$$

$$= 1 - \frac{\sum_{L\in N_{ij}}\prod_{ab\in L\setminus\{ij\}}\exp\big(\beta_t(1-s_{ab,(t)})\big)d_L}{\sum_{L\in N_{ij}}\prod_{ab\in L\setminus\{ij\}}\exp\big(\beta_t(1-s_{ab,(t)})\big)}.$$

$$= 1 - \frac{\sum_{L\in N_{ij}}\exp((l-1)\beta_t)\prod_{ab\in L\setminus\{ij\}}\exp\big(-\beta_t s_{ab,(t)}\big)d_L}{\sum_{L\in N_{ij}}\exp((l-1)\beta_t)\prod_{ab\in L\setminus\{ij\}}\exp\big(-\beta_t s_{ab,(t)}\big)}$$

$$= 1 - \frac{\sum_{L\in N_{ij}}\prod_{ab\in L\setminus\{ij\}}\exp\big(-\beta_t s_{ab,(t)}\big)d_L}{\sum_{L\in N_{ij}}\prod_{ab\in L\setminus\{ij\}}\exp\big(-\beta_t s_{ab,(t)}\big)} = 1 - s_{ij,(t+1)}.$$

Consequently, for all $ij\in E$ and $t\geq 0$, $\boldsymbol{A}_{\text{init},(t)}(i,j)=1-s_{ij,(t)}$, and thus Algorithm 1 is equivalent to the original CEMP.

**Remark A.1.** *The equivalence between CEMP and Algorithm 1 is not restricted to permutations. Indeed, our arguments apply to any group that can be represented as a subgroup of the orthogonal group $O(m)$, where*

*we assign to any the representations of the elements $\boldsymbol{X}$ and $\boldsymbol{Y}$ the semimetric $d(\boldsymbol{X},\boldsymbol{Y})=\|\boldsymbol{X}-\boldsymbol{Y}\|_F^2/2m$. For permutation synchronization, this is a metric and thus the theory of CEMP directly extends. For other groups, this semimetric satisfies a relaxed triangle inequality (with constant 2). Thus one may still extend the theory of CEMP, but with weaker estimates.*

## A.3  Proof of Theorem 5.2

We describe the proof in two different sections. In §A.3.1 we prove the first part of the theorem that guarantees sufficiently near recovery by CEMP under the superspreader model. In §A.3.2 we verify that other least squares methods generally do not succeed with recovery under our nonuniform setting.

### A.3.1  A theoretical guarantee for CEMP in the nonuniform case.

For each $i \in N$ and $ij \in E$, we recall the definitions: $N(i) = \{j \in [n] : ij \in E\}$ and $N(ij) = \{k \in [n] : ik, jk \in E\}$. Similarly, for any $ij \in E$, we recall that $N_g(ij) = \{k \in N(ij) : k \neq i,j \text{ and } ik, jk \in E_g\}$ and we also define $N_b(ij) = N(ij) \setminus N_g(ij)$. We will use the following Chernoff bound. For i.i.d. Bernoulli random variables $\{X_i\}_{i=1}^M$ with means $\mu$ and any $0 < \eta < 1$,

$$\Pr\left(\left|\sum_{i=1}^M X_i - M\mu\right| > \eta M\mu\right) < 2e^{-\frac{\eta^2}{3}\mu M}. \tag{24}$$

We first show that the following deterministic conditions hold with high probability under the assumptions of our model:

$$\frac{1}{2}np \leq N(i) \leq 2np \quad \text{for any } i \in [n], \tag{25}$$

$$\frac{1}{2}np^2 \leq N(ij) \leq 2np^2 \quad \text{for any } ij \in E, \tag{26}$$

$$\frac{1}{2}|N(i_0j)|\varepsilon \leq N_g(i_0j) \leq 2|N(i_0j)|\varepsilon \quad \text{for any } j \in N(i_0). \tag{27}$$

Indeed, since $\mathbf{1}_{\{j \in N(i)\}}$, $\mathbf{1}_{\{k \in N(ij)\}}$, $\mathbf{1}_{\{k \in N_g(i_0j)\}}$ are all Bernoulli random variables with mean $p, p^2, \varepsilon$, respectively, by first applying the Chernoff bound (24) to the above Bernoulli random variables with $M = n, n, |N(i_0j)|$ and then the union bound over $i \in [n]$, $ij \in E$ and $j \in N(i_0)$ we obtain that (25)-(27) hold with probability at least

$$1 - 2n\exp(\Omega(np)) - 2|E|\exp(\Omega(np^2)) - 2|N(i_0)|\exp(\Omega(np^2\varepsilon)) \tag{28}$$

$$= 1 - 2n\exp(\Omega(np)) - 4np^2\exp(\Omega(np^2)) - 4np\exp(\Omega(np^2\varepsilon)). \tag{29}$$

Indeed, the probability in (29) is high given our assumption $n = \Omega(1/(p^2\mu^2\varepsilon^2))$. We next show that the theorem holds with high probability given (25)-(27).

We recall that $\boldsymbol{A}_{\text{init},(0)} = \frac{1}{m}\langle \boldsymbol{S}_{\text{init},(0)}^2 \oslash (\boldsymbol{W}_{\text{init},(0)}^2 \otimes \mathbf{1}_m), \tilde{\boldsymbol{X}}\rangle_{\text{block}}$, where $\boldsymbol{S}_{\text{init},(t)} = (\boldsymbol{W}_{\text{init},(t)} \otimes \mathbf{1}_m) \odot \tilde{\boldsymbol{X}}$. We note that for $ij \in E$

$$\boldsymbol{A}_{\text{init},(0)}(i,j) = \langle \tilde{\boldsymbol{X}}_{ij}, \sum_{k \in N(ij)} \tilde{\boldsymbol{X}}_{ik}\tilde{\boldsymbol{X}}_{kj}\rangle/(m|N(ij)|). \tag{30}$$

We further note that for $i_0j \in E_b$ and $i_0k \in E_b$

$$a_{i_0jk}^{bb} := \langle \tilde{\boldsymbol{X}}_{i_0j}, \tilde{\boldsymbol{X}}_{i_0k}\tilde{\boldsymbol{X}}_{kj}\rangle/m = \langle \tilde{\boldsymbol{X}}_{i_0j}, \tilde{\boldsymbol{X}}_{i_0k}\boldsymbol{P}_k^*\boldsymbol{P}_j^{*\intercal}\rangle/m; \tag{31}$$

for $i_0j \in E_b$, and $i_0k \in E_g$

$$a_{i_0jk}^{bg} := \langle \tilde{\boldsymbol{X}}_{i_0j}, \tilde{\boldsymbol{X}}_{i_0k}\tilde{\boldsymbol{X}}_{kj}\rangle/m = \langle \tilde{\boldsymbol{X}}_{i_0j}, \boldsymbol{P}_{i_0}^*\boldsymbol{P}_k^{*\intercal}\boldsymbol{P}_k^*\boldsymbol{P}_j^{*\intercal}\rangle/m = \langle \tilde{\boldsymbol{X}}_{i_0j}, \tilde{\boldsymbol{P}}_{i_0}^*\boldsymbol{P}_j^{*\intercal}\rangle/m; \tag{32}$$

for $i_0j \in E_g$, and $i_0k \in E_b$

$$a_{i_0jk}^{gb} := \langle \tilde{\boldsymbol{X}}_{i_0j}, \tilde{\boldsymbol{X}}_{i_0k}\tilde{\boldsymbol{X}}_{kj}\rangle/m = \langle \boldsymbol{P}_{i_0}^*\boldsymbol{P}_j^{*\intercal}, \tilde{\boldsymbol{X}}_{i_0k}\boldsymbol{P}_k^*\boldsymbol{P}_j^{*\intercal}\rangle/m = \langle \tilde{\boldsymbol{X}}_{i_0k}, \tilde{\boldsymbol{P}}_{i_0}^*\boldsymbol{P}_k^{*\intercal}\rangle/m; \tag{33}$$

and for $i_0j \in E_g$, and $i_0k \in E_g$

$$a_{i_0jk}^{gg} = 1. \tag{34}$$

We denote the expectations of $a_{i_0jk}^{bb}$, $a_{i_0jk}^{bg}$, $a_{i_0jk}^{gb}$ by $\mu_{bb}$, $\mu_{bg}$ and $\mu_{gb}$, respectively. We use this notation and the above equations to estimate $\mathbb{E}\big(\boldsymbol{A}_{\text{init},(0)}(i_0,j)|i_0j \in E_b\big)$ and $\mathbb{E}\big(\boldsymbol{A}_{\text{init},(0)}(i_0,j)|i_0j \in E_g\big)$. For this purpose, we note that for $i_0j \in E_b$ and $k \neq j,i_0$, there are $(1-\varepsilon)|N(i_0)|-1$ edges $i_0k \in E_b$ and $\varepsilon|N(i_0)|$ edges $i_0k \in E_g$. Similarly, for $i_0j \in E_g$ and $k \neq j,i_0$, there are $(1-\varepsilon)|N(i_0)|$ edges $i_0k \in E_b$ and $\varepsilon|N(i_0)|-1$ edges $i_0k \in E_g$. Combining (31)-(34) with these observations, we conclude that

$$\mathbb{E}\big(\boldsymbol{A}_{\mathrm{init},(0)}(i_0,j)|i_0 j\in E_g\big)=\mathbb{E}\left(\frac{1}{|N(i_0 j)|}\left(\sum_{k\in N_b(i_0 j)}a^{gb}_{i_0 jk}+\sum_{k\in N_g(i_0 j)}a^{gg}_{i_0 jk}\right)\bigg| i_0 j\in E_g\right)$$

$$=(1-\varepsilon)\mu_{gb}+\varepsilon \tag{35}$$

and

$$\mathbb{E}\big(\boldsymbol{A}_{\mathrm{init},(0)}(i_0,j)|i_0 j\in E_b\big)=\mathbb{E}\left(\frac{1}{|N(i_0 j)|}\left(\sum_{k\in N_b(i_0 j)}a^{bb}_{i_0 jk}+\sum_{k\in N_g(i_0 j)}a^{bg}_{i_0 jk}\right)\bigg| i_0 j\in E_b\right)$$

$$=(1-\varepsilon)\mu_{bb}+\varepsilon\mu_{bg}. \tag{36}$$

Next we prove that

$$\mu_{bb}\le\mu_{gb}=\mu_{bg}=1-\mu. \tag{37}$$

Note that (13) and (22) imply that

$$\mu=\mathbb{E}(m-\langle\tilde{\boldsymbol{X}}_{i_0 j},\boldsymbol{X}^*_{i_0 j}\rangle\,|\,j\in N_b(i_0))/m\le\mathbb{E}(m-\langle\tilde{\boldsymbol{X}}_{ki_0}\tilde{\boldsymbol{X}}_{i_0 j},\boldsymbol{X}^*_{kj}\rangle\,|\,j,k\in N_b(i_0))/m.$$

Applying this equation, we conclude (37) as follows:

$$\mu_{gb}=\mathbb{E}(\langle\tilde{\boldsymbol{X}}_{i_0 j},\boldsymbol{X}^*_{i_0 j}\rangle\,|\,j\in N_b(i_0))/m\ge\mathbb{E}(\langle\tilde{\boldsymbol{X}}_{ki_0}\tilde{\boldsymbol{X}}_{i_0 j},\boldsymbol{X}^*_{kj}\rangle\,|\,j,k\in N_b(i_0))/m$$

$$=\mathbb{E}(\langle\tilde{\boldsymbol{X}}_{i_0 j},\tilde{\boldsymbol{X}}_{i_0 k}\boldsymbol{X}^*_{kj}\rangle\,|\,j,k\in N_b(i_0))/m=\mathbb{E}(\langle\tilde{\boldsymbol{X}}_{i_0 j},\tilde{\boldsymbol{X}}_{i_0 k}\boldsymbol{P}^*_k\boldsymbol{P}^{*\intercal}_j\rangle\,|\,j,k\in N_b(i_0))/m=\mu_{bb}.$$

Note that the combination of (35), (36) and (37) yields

$$\mathbb{E}\big(\boldsymbol{A}_{\mathrm{init},(0)}(i_0,j)|i_0 j\in E_g\big)-\mathbb{E}\big(\boldsymbol{A}_{\mathrm{init},(0)}(i_0,k)|i_0 k\in E_b\big)\ge(1-\mu_{bg})\varepsilon=\mu\varepsilon. \tag{38}$$

We also note that for any $jk\in E$, where $i_0\in N(jk)$, the cycle $jki_0$ is the only cycle that contain $jk$ whose edges may belong to $E_b$. We thus use (30), then the fact that for $\boldsymbol{X},\boldsymbol{Y}\in\mathscr{P}_m$, $\langle\boldsymbol{X},\boldsymbol{Y}\rangle\ge 0$ together with the latter observation. At last, we use the fact that $\langle\boldsymbol{X}^*_{jk},\boldsymbol{X}^*_{ji}\boldsymbol{X}_{ik}\rangle=\langle\boldsymbol{X}^*_{jk},\boldsymbol{X}^*_{jk}\rangle=m$ to conclude that for any $jk\in E$

$$\boldsymbol{A}_{\mathrm{init},(0)}(j,k)=\frac{1}{m|N(jk)|}\left(\langle\tilde{\boldsymbol{X}}_{jk},\tilde{\boldsymbol{X}}_{ji_0}\tilde{\boldsymbol{X}}_{i_0 k}\rangle+\sum_{i\in N(jk)\backslash i_0}\langle\tilde{\boldsymbol{X}}_{jk},\tilde{\boldsymbol{X}}_{ji}\tilde{\boldsymbol{X}}_{ik}\rangle\right)$$

$$\ge\frac{1}{m|N(jk)|}\sum_{i\in N(jk)\backslash i_0}\langle\boldsymbol{X}^*_{jk},\boldsymbol{X}^*_{ji}\boldsymbol{X}^*_{ik}\rangle=\frac{1}{m|N(jk)|}(|N(jk)|-1)m=1-\frac{1}{|N(jk)|}\ge 1-\frac{2}{np^2},$$

and consequently

$$\max_{jk,j'k'\in E}|\boldsymbol{A}_{\mathrm{init},(0)}(j,k)-\boldsymbol{A}_{\mathrm{init},(0)}(j',k')|\le\frac{2}{np^2}. \tag{39}$$

We note that for the given $i_0\in[n]$, $j\in N(i_0)$ and $k\in N(i_0 j)$: $a^{bb}_{i_0 jk}$, $a^{bg}_{i_0 jk}$, $a^{gb}_{i_0 jk}$ are all independent random variables $\in[0,1]$. Therefore, application of Hoeffding's inequality and the assumption that $n=\Omega(1/(p^2\mu^2\varepsilon^2))$ yields for $j\in N_g(i_0)$

$$\Pr\left(\boldsymbol{A}_{\mathrm{init},(0)}(i_0,j)\le\mathbb{E}\boldsymbol{A}_{\mathrm{init},(0)}(i_0,j)-\frac{\mu\varepsilon}{4}+\frac{1}{np^2}\right)$$

$$<\exp\left(-\Omega\left(np^2\left(\frac{\mu\varepsilon}{4}-\frac{1}{np^2}\right)^2\right)\right)=\exp\left(-\Omega\big(np^2\mu^2\varepsilon^2\big)\right) \tag{40}$$

and for $k\in N_b(i_0)$

$$\Pr\left(\boldsymbol{A}_{\mathrm{init},(0)}(i_0,k)\ge\mathbb{E}\boldsymbol{A}_{\mathrm{init},(0)}(i_0,k)+\frac{\mu\varepsilon}{4}-\frac{1}{np^2}\right)$$

$$<\exp\left(-\Omega\left(np^2\left(\frac{\mu\varepsilon}{4}-\frac{1}{np^2}\right)^2\right)\right)=\exp\left(-\Omega\big(np^2\mu^2\varepsilon^2\big)\right). \tag{41}$$

Taking a union bound over $j\in N_g(i_0)$, while using (40), and another union bound over $k\in N_b(i_0)$, while using (41), result in

$$\Pr\left(\min_{j\in N_g(i_0)}\boldsymbol{A}_{\mathrm{init},(0)}(i_0,j)\ge\mathbb{E}\big(\boldsymbol{A}_{\mathrm{init},(0)}(i_0,j)|i_0 j\in E_g\big)-\frac{\mu\varepsilon}{4}+\frac{1}{np^2}\right)>1-2np\exp\big(-\Omega\big(np^2\mu^2\varepsilon^2\big)\big)$$

and

$$\Pr\left(\max_{k\in N_b(i_0)}\boldsymbol{A}_{\mathrm{init},(0)}(i_0,k)\le\mathbb{E}\big(\boldsymbol{A}_{\mathrm{init},(0)}(i_0,k)|i_0 k\in E_b\big)+\frac{\mu\varepsilon}{4}-\frac{1}{np^2}\right)>1-2np\exp\big(-\Omega\big(np^2\mu^2\varepsilon^2\big)\big).$$

Combining the above two equations and then applying (38) we obtain that

$$\Pr\Bigg(\min_{j\in N_g(i_0)}\boldsymbol{A}_{\text{init},(0)}(i_0,j)>\max_{k\in N_b(i_0)}\boldsymbol{A}_{\text{init},(0)}(i_0,k)+$$

$$\mathbb{E}(\boldsymbol{A}_{\text{init},(0)}(i_0,j)|i_0j\in E_g)-\mathbb{E}(\boldsymbol{A}_{\text{init},(0)}(i_0,k)|i_0k\in E_b)-\frac{\mu\varepsilon}{2}+\frac{2}{np^2}\Bigg) \tag{42}$$

$$\geq\Pr\Bigg(\min_{j\in N_g(i_0)}\boldsymbol{A}_{\text{init},(0)}(i_0,j)>\max_{k\in N_b(i_0)}\boldsymbol{A}_{\text{init},(0)}(i_0,k)+\frac{\mu\varepsilon}{2}+\frac{2}{np^2}\Bigg)$$

$$>1-4np\exp\big(-\Omega\big(np^2\mu^2\varepsilon^2\big)\big).$$

The combination of (39) and (42) yields for any $j\neq i_0$

$$\Pr\Bigg(\min_{k\in N_g(i_0j)}(\boldsymbol{A}_{\text{init},(0)}(i_0,k)+\boldsymbol{A}_{\text{init},(0)}(k,j))>\max_{k\in N_b(i_0j)}(\boldsymbol{A}_{\text{init},(0)}(i_0,k)+\boldsymbol{A}_{\text{init},(0)}(k,j))+\frac{\mu}{2}\varepsilon\Bigg)$$

$$>1-4np\exp\big(-\Omega\big(np^2\mu^2\varepsilon^2\big)\big).$$

Recall that for $ij\in E$, $\boldsymbol{W}_{\text{init},(1)}(i,j)=\exp(\beta_0\boldsymbol{A}_{\text{init},(0)}(i,j))$. In view of this equality and the above equation, we conclude that for any $j\neq i_0$

$$\min_{k\in N_g(i_0j)}\boldsymbol{W}_{\text{init},(1)}(i_0,k)\boldsymbol{W}_{\text{init},(1)}(k,j)\geq\max_{k\in N_b(i_0j)}\boldsymbol{W}_{\text{init},(1)}(i_0,k)\boldsymbol{W}_{\text{init},(1)}(k,j)e^{\beta_0\mu\varepsilon/2}$$

$$\text{with probability at least }1-4np\exp\big(-\Omega\big(np^2\mu^2\varepsilon^2\big)\big). \tag{43}$$

Using this inequality, we establish the desired upper bound of $\|\boldsymbol{A}_{\text{init},(1)}-\boldsymbol{A}^*\|_\infty$ in the following three complementary cases.

**Case 1**: Edge $ij\in E$ is incident to node $i_0$. That is, without loss of generality, the edge $ij$ is of the form $i_0j$ for $j\in[n]\setminus\{i_0\}$. In this case, by assumption (27),

$$\frac{|N_g(i_0j)|}{|N_b(i_0j)|}\geq\frac{|N(i_0j)|\varepsilon/2}{|N(i_0j)|(1-\varepsilon/2)}=\frac{\varepsilon}{2-\varepsilon}. \tag{44}$$

Combining the definition of $\boldsymbol{A}_{\text{init},(1)}$, the fact that $|\langle\tilde{\boldsymbol{X}}_{i_0j},\tilde{\boldsymbol{X}}_{i_0k}\tilde{\boldsymbol{X}}_{kj}\rangle/m-\boldsymbol{A}^*(i_0,j)|\leq1$ (as it is an absolute value of a difference of two numbers in $[0,1]$) as well as (43) and (44), we obtain that with the probability indicated in (43)

$$|\boldsymbol{A}_{\text{init},(1)}(i_0,j)-\boldsymbol{A}^*(i_0,j)|$$

$$=\left|\frac{\sum_{k\in N(i_0j)}\boldsymbol{W}_{\text{init},(1)}(i_0,k)\boldsymbol{W}_{\text{init},(1)}(k,j)\Big(\langle\tilde{\boldsymbol{X}}_{i_0j},\tilde{\boldsymbol{X}}_{i_0k}\tilde{\boldsymbol{X}}_{kj}\rangle/m\Big)}{\sum_{k\in N(i_0j)}\boldsymbol{W}_{\text{init},(1)}(i_0,k)\boldsymbol{W}_{\text{init},(1)}(k,j)}-\boldsymbol{A}^*(i_0,j)\right|$$

$$\leq\frac{\sum_{k\in N(i_0j)}\boldsymbol{W}_{\text{init},(1)}(i_0,k)\boldsymbol{W}_{\text{init},(1)}(k,j)|\langle\tilde{\boldsymbol{X}}_{i_0j},\tilde{\boldsymbol{X}}_{i_0k}\tilde{\boldsymbol{X}}_{kj}\rangle/m-\boldsymbol{A}^*(i_0,j)|}{\sum_{k\in N(i_0j)}\boldsymbol{W}_{\text{init},(1)}(i_0,k)\boldsymbol{W}_{\text{init},(1)}(k,j)} \tag{45}$$

$$\leq\frac{\sum_{k\in N_b(i_0j)}\boldsymbol{W}_{\text{init},(1)}(i_0,k)\boldsymbol{W}_{\text{init},(1)}(k,j)}{\sum_{k\in N_b(i_0j)}\boldsymbol{W}_{\text{init},(1)}(i_0,k)\boldsymbol{W}_{\text{init},(1)}(k,j)+\sum_{k\in N_g(i_0j)}\boldsymbol{W}_{\text{init},(1)}(i_0,k)\boldsymbol{W}_{\text{init},(1)}(k,j)}$$

$$=\frac{1}{1+\frac{\sum_{k\in N_g(i_0j)}\boldsymbol{W}_{\text{init},(1)}(i_0,k)\boldsymbol{W}_{\text{init},(1)}(k,j)}{\sum_{k\in N_b(i_0j)}\boldsymbol{W}_{\text{init},(1)}(i_0,k)\boldsymbol{W}_{\text{init},(1)}(k,j)}}\leq\frac{1}{1+\frac{|N_g(i_0j)|}{|N_b(i_0j)|}e^{\beta_0\mu\varepsilon/2}}\leq\frac{1}{1+\frac{\varepsilon}{2-\varepsilon}e^{\beta_0\mu\varepsilon/2}}.$$

**Case 2**: Edge $jk$ is not incident to $i_0$ and $i_0\in N_b(jk)$. That is, we assume that $j$ and $k$ are in $[n]\setminus\{i_0\}$ and at least one of them is in $N_b(i_0)$. In this case,

$$N_b(jk)=\{i_0\}\quad\text{and}\quad N_g(jk)=N(jk)\setminus\{i_0\}$$

and consequently

$$|N_b(jk)|=1\quad\text{and}\quad|N_g(jk)|=|N(jk)|-1.$$

Following the same arguments deriving (45), but using the above two equations (instead of (44)), we obtain that with the probability indicated in (43)

$$|\boldsymbol{A}_{\text{init},(1)}(j,k)-\boldsymbol{A}^*(j,k)|\leq\frac{\boldsymbol{W}_{\text{init},(1)}(j,i_0)\boldsymbol{W}_{\text{init},(1)}(i_0,k)}{\boldsymbol{W}_{\text{init},(1)}(j,i_0)\boldsymbol{W}_{\text{init},(1)}(i_0,k)+\sum_{l\in N(jk)\setminus\{i_0\}}\boldsymbol{W}_{\text{init},(1)}(j,l)\boldsymbol{W}_{\text{init},(1)}(l,k)}$$

$$=\frac{1}{1+\frac{\sum_{l\in N(jk)\setminus\{i_0\}}\boldsymbol{W}_{\text{init},(1)}(j,l)\boldsymbol{W}_{\text{init},(1)}(l,k)}{\boldsymbol{W}_{\text{init},(1)}(j,i_0)\boldsymbol{W}_{\text{init},(1)}(i_0,k)}}\leq\frac{1}{1+(|N(jk)|-1)e^{\beta_0\mu\varepsilon/2}}.$$

**Case 3**: Edge $jk$ is not incident to $i_0$ and $i_0\in N_g(jk)$. That is, we assume that both $j$ and $k$ are in $N_g(i_0)$. Note that in this case, all 3-cycles containing $jk$ are uncorrupted. That is

$$N_b(jk)=\emptyset\quad\text{and}\quad N_g(jk)=N(jk).$$

Consequently, $\boldsymbol{A}_{\text{init},(1)}(j,k)=\boldsymbol{A}^*(j,k)=1$ and thus
$$|\boldsymbol{A}_{\text{init},(1)}(j,k)-\boldsymbol{A}^*(j,k)|=0.$$
Combining all the above three cases, for all $jk\in E$
$$|\boldsymbol{A}_{\text{init},(1)}(j,k)-\boldsymbol{A}^*(j,k)|\leq\frac{1}{1+\frac{\varepsilon}{2-\varepsilon}e^{\beta_0\mu\varepsilon/2}}.$$
with probability at least $1-4np\exp\left(-\Omega\left(np^2\mu^2\varepsilon^2\right)\right)$. Since $n=\Omega(1/(p^2\mu^2\varepsilon^2))$ this probability is sufficiently large. Note that the only free parameter in the right hand side of the above inequality is $\beta_0$. Thus one can apply an aggressive reweighting with very large $\beta_0$ and guarantee in this special case near exact recovery for CEMP. The only restriction of Theorem (5.2) on $\{\tilde{\boldsymbol{X}}_{ij}\}_{ij\in E}$ and $\{\boldsymbol{P}_i^*\}_{i\in[n]}$ is the condition:
$$\mathbb{E}(\|\tilde{\boldsymbol{X}}_{i_0j}-\boldsymbol{X}_{i_0j}^*\|_F^2\,|\,j\in N_b(i_0))\leq\mathbb{E}(\|\tilde{\boldsymbol{X}}_{ki_0}\tilde{\boldsymbol{X}}_{i_0j}-\boldsymbol{X}_{kj}^*\|_F^2\,|\,j,k\in N_b(i_0)).$$
We note that since
$$\|\tilde{\boldsymbol{X}}_{i_0j}-\boldsymbol{X}_{i_0j}^*\|_F^2=\|\boldsymbol{X}_{ki_0}^*\tilde{\boldsymbol{X}}_{i_0j}\boldsymbol{X}_{jk}^*-\boldsymbol{I}_m\|_F^2\quad\text{and}\quad\|\tilde{\boldsymbol{X}}_{ki_0}\tilde{\boldsymbol{X}}_{i_0j}-\boldsymbol{X}_{kj}^*\|_F^2=\|\tilde{\boldsymbol{X}}_{ki_0}\tilde{\boldsymbol{X}}_{i_0j}\boldsymbol{X}_{jk}^*-\boldsymbol{I}_m\|_F^2$$
this condition is equivalent to
$$\mathbb{E}\left(\|\boldsymbol{X}_{ki_0}^*\tilde{\boldsymbol{X}}_{i_0j}\boldsymbol{X}_{jk}^*-\boldsymbol{I}_m\|_F^2\,|\,j\in N_b(i_0)\right)\leq\mathbb{E}\left(\|\tilde{\boldsymbol{X}}_{ki_0}\tilde{\boldsymbol{X}}_{i_0j}\boldsymbol{X}_{jk}^*-\boldsymbol{I}_m\|_F^2\,|\,j,k\in N_b(i_0)\right).$$
Both sides of the inequality contain conditional expectations of the cycle inconsistency of the 3 cycle $i_0jk$. In the LHS it is condition on the edge $i_0j$ being corrupted, where in the RHS it is conditioned on both edges $i_0j$ and $i_0k$ being corrupted. That is, the above condition means that when the number of corrupted edges in a 3-cycle is enlarged from 1 to 2, then the cycle inconsistency increases on average.

### A.3.2  Failure cases of least squares methods

We demonstrate some failure cases of least squares methods for permutation synchronization under the superspreader model. In view of (27), we assume that $(1-2\varepsilon)$-fraction of $i_0j\in E$ is corrupted.

We start with considering PPM. In view of (4), the PPM iteration at node $i_0$ is
$$\boldsymbol{P}_{i_0,(t+1)}=\underset{\boldsymbol{P}_{i_0}\in\mathscr{P}_m}{\text{argmax}}\left\langle\boldsymbol{P}_{i_0},\frac{1}{|N(i_0)|+1}\sum_{j\in[n]}\tilde{\boldsymbol{X}}_{i_0j}\boldsymbol{P}_{j,(t)}\right\rangle.$$
The following proposition demonstrates failure cases of PPM. It uses the notation $\boldsymbol{Q}=\sum_{j\in N_b(i_0)}\tilde{\boldsymbol{X}}_{i_0j}\boldsymbol{P}_j^*/|N_b(i_0)|$.

**Proposition A.2.** *If there exist $\boldsymbol{P}_{crpt}\neq\boldsymbol{P}_{i_0}^*$ and $\varepsilon_0<1$ such that $2\varepsilon\sqrt{2m}+(1-2\varepsilon)\varepsilon_0<1$ and*
$$\|\boldsymbol{Q}-\boldsymbol{P}_{crpt}\|_F<\varepsilon_0,\tag{46}$$
*then*
$$\boldsymbol{P}_{crpt}=\underset{\boldsymbol{P}_{i_0}\in\mathscr{P}_m}{\text{argmax}}\left\langle\boldsymbol{P}_{i_0},\frac{1}{|N(i_0)|+1}\sum_{j\in[n]}\tilde{\boldsymbol{X}}_{i_0j}\boldsymbol{P}_j^*\right\rangle\tag{47}$$
*and thus PPM cannot recover $\boldsymbol{P}_{i_0}^*$.*

Before we prove this proposition, we clarify it. It states that if the average of $\tilde{\boldsymbol{X}}_{i_0j}\boldsymbol{P}_j^*$ over $j\in N_b(i_0)$ concentrates around a certain permutation matrix, which is different than $\boldsymbol{P}_{i_0}^*$, and $\varepsilon$ is sufficiently small, then PPM fails to recover the ground-truth permutations. By law of large numbers, the condition $\|\boldsymbol{Q}-\boldsymbol{P}_{crpt}\|_F<\varepsilon_0$ is satisfied when
$$\left\|\mathbb{E}(\tilde{\boldsymbol{X}}_{i_0j}\boldsymbol{P}_j^*)-\boldsymbol{P}_{crpt}\right\|_F<\varepsilon_0/2\tag{48}$$
and $|N(i_0)|$ is sufficiently large. The LAC model described in §6.1 represents this setting. Indeed, in this case $\boldsymbol{P}_{crpt}$ is the identity matrix $\boldsymbol{I}_m$ and $\tilde{\boldsymbol{X}}_{i_0j}\boldsymbol{P}_j^*$ randomly permutes 3 columns of the identity. In this case, $E(\tilde{\boldsymbol{X}}_{i_0j}\boldsymbol{P}_j^*)(i,i)=(m-3)/m$ for $i\in[m]$ and $E(\tilde{\boldsymbol{X}}_{i_0j}\boldsymbol{P}_j^*)(i,j)=3/(m(m-1))$ for $i,j\in[m]$, where $i\neq j$. We have tested this case with $m=10$, where we have violated the condition $2\varepsilon\sqrt{2m}+(1-2\varepsilon)\varepsilon_0<1$. Nevertheless, we have still seen a clear advantage of IRGCL, which uses CEMP, over PPM.

We believe that condition (13) for CEMP is less restrictive than (48). Nevertheless, we point out that in the deterministic case when $\tilde{\boldsymbol{X}}_{i_0j}\boldsymbol{P}_j^*=\boldsymbol{P}_{crpt}$ for all $j\in N_b(i_0)$, then both CEMP and PPM fail. We first note that the RHS of (13) is 0, so the proposition does not hold for CEMP. We also note that in this scenario the problem of exact recovery is ill-posed as no algorithm can recover $\boldsymbol{P}_{i_0}^*$. Indeed, setting $\tilde{\boldsymbol{X}}_{i_0j}\boldsymbol{P}_j^*=\boldsymbol{P}_{crpt}$ is equivalent to corrupting $\boldsymbol{X}_{ij}^*=\boldsymbol{P}_i^*\boldsymbol{P}_j^{*\mathsf{T}}$ as follows: $\tilde{\boldsymbol{X}}_{ij}:=\boldsymbol{P}_{crpt}\boldsymbol{P}_j^{*\mathsf{T}}$ and thus replacing the underlying ground-truth permutation $\boldsymbol{P}_i^*$ by $\boldsymbol{P}_{crpt}$. Anyway, Proposition A.2 assumes a much broader scenario than this special deterministic example.

*Proof.* We note that for any $\boldsymbol{P}_{i_0} \in \mathscr{P}_m$

$$\left\langle \boldsymbol{P}_{i_0}, \frac{1}{|N(i_0)|+1} \sum_{j\in[n]} \tilde{\boldsymbol{X}}_{i_0 j} \boldsymbol{P}_j^* \right\rangle = \left\langle \boldsymbol{P}_{i_0}, \frac{1}{|N(i_0)|+1} \left( \boldsymbol{P}_{i_0}^* + \sum_{j\in N_g(i_0)} \boldsymbol{X}_{i_0 j}^* \boldsymbol{P}_j^* + \sum_{j\in N_b(i_0)} \tilde{\boldsymbol{X}}_{i_0 j} \boldsymbol{P}_j^* \right) \right\rangle$$

$$= \left\langle \boldsymbol{P}_{i_0}, \frac{(|N_g(i_0)|+1)\boldsymbol{P}_{i_0}^* + |N_b(i_0)|\boldsymbol{Q}}{|N(i_0)|+1} \right\rangle.$$

Therefore, it is sufficient to prove that

$$\boldsymbol{P}_{\text{crpt}} = \underset{\boldsymbol{P}_{i_0}\in\mathscr{P}_m}{\operatorname{argmax}} \left\langle \boldsymbol{P}_{i_0}, \frac{1}{|N(i_0)|+1} \sum_{j\in[n]} \tilde{\boldsymbol{X}}_{i_0 j} \boldsymbol{P}_j^* \right\rangle = \underset{\boldsymbol{P}_{i_0}\in\mathscr{P}_m}{\operatorname{argmax}} \left\langle \boldsymbol{P}_{i_0}, \hat{\boldsymbol{P}}_{i_0} \right\rangle, \tag{49}$$

where $\boldsymbol{P}_{\text{crpt}} \neq \boldsymbol{P}_{i_0}^*$ and $\hat{\boldsymbol{P}}_{i_0} = ((|N_g(i_0)|+1)\boldsymbol{P}_{i_0}^* + |N_b(i_0)|\boldsymbol{Q})/(|N(i_0)|+1)$. Using basic algebraic relationships and at last applying together the conditions $2\varepsilon\sqrt{2m}+(1-2\varepsilon)\varepsilon_0 < 1$ and $\varepsilon_0 < 1$, the fact that $\|\boldsymbol{X}-\boldsymbol{Y}\|_F^2/2m \in [0,1]$ for any $\boldsymbol{X},\boldsymbol{Y}\in\mathscr{P}_m$ and (46), we obtain that for $|N(i_0)|$ sufficiently large

$$\|\hat{\boldsymbol{P}}_{i_0} - \boldsymbol{P}_{\text{crpt}}\|_F = \left\| \frac{(|N_g(i_0)|+1)\boldsymbol{P}_{i_0}^* + |N_b(i_0)|\boldsymbol{Q}}{|N(i_0)|+1} - \boldsymbol{P}_{\text{crpt}} \right\|_F$$

$$= \left\| \frac{|N_g(i_0)|+1}{|N(i_0)|+1}(\boldsymbol{P}_{i_0}^* - \boldsymbol{P}_{\text{crpt}}) + \frac{|N_b(i_0)|}{|N(i_0)|+1}(\boldsymbol{Q} - \boldsymbol{P}_{\text{crpt}}) \right\|_F$$

$$\leq \left\| \frac{|N_g(i_0)|+1}{|N(i_0)|+1}(\boldsymbol{P}_{i_0}^* - \boldsymbol{P}_{\text{crpt}}) \right\|_F + \left\| \frac{|N_b(i_0)|}{|N(i_0)|+1}(\boldsymbol{Q} - \boldsymbol{P}_{\text{crpt}}) \right\|_F$$

$$\leq 2\varepsilon\sqrt{2m}+(1-2\varepsilon)\varepsilon_0 < 1. \tag{50}$$

By combining (50) and the fact that $\|\boldsymbol{X}-\boldsymbol{Y}\|_F \geq 2$ for $\boldsymbol{X}\neq\boldsymbol{Y}\in\mathscr{P}_m$, we obtain that

$$\|\hat{\boldsymbol{P}}_{i_0} - \boldsymbol{P}_{\text{crpt}}\|_F < \frac{1}{2} \min_{\boldsymbol{P}'\in\mathscr{P}_m, \boldsymbol{P}'\neq\boldsymbol{P}_{\text{crpt}}} \|\boldsymbol{P}' - \boldsymbol{P}_{\text{crpt}}\|_F.$$

Consequently, we conclude (49) and thus the auxiliary proposition as follows

$$\boldsymbol{P}_{\text{crpt}} = \underset{\boldsymbol{P}_{i_0}\in\mathscr{P}_m}{\operatorname{argmin}} \|\boldsymbol{P}_{i_0} - \hat{\boldsymbol{P}}_{i_0}\|_F = \underset{\boldsymbol{P}_{i_0}\in\mathscr{P}_m}{\operatorname{argmax}} \left\langle \boldsymbol{P}_{i_0}, \hat{\boldsymbol{P}}_{i_0} \right\rangle.$$

$\square$

The argument for failure of general least squares methods is more delicate. Using the above rigorous argument for PPM, we provide some intuition why least methods can fail. We note that such methods aim to solve

$$\max_{\{\boldsymbol{P}_i\}_{i\in[n]}\subset\mathscr{P}_m} \sum_{j\in[n]}\sum_{k\in[n]} \left\langle \boldsymbol{P}_j\boldsymbol{P}_k^\intercal, \tilde{\boldsymbol{X}}_{jk} \right\rangle. \tag{51}$$

We rewrite the objective function of (51) as follows

$$\sum_{j\in[n]}\sum_{k\in[n]} \left\langle \boldsymbol{P}_j\boldsymbol{P}_k^\intercal, \tilde{\boldsymbol{X}}_{jk} \right\rangle$$

$$= \left\langle \boldsymbol{P}_{i_0}\boldsymbol{P}_{i_0}^\intercal, \boldsymbol{I}_m \right\rangle + \sum_{j\neq i_0} \left\langle \boldsymbol{P}_j\boldsymbol{P}_{i_0}^\intercal, \tilde{\boldsymbol{X}}_{ji_0} \right\rangle + \sum_{k\neq i_0} \left\langle \boldsymbol{P}_{i_0}\boldsymbol{P}_k^\intercal, \tilde{\boldsymbol{X}}_{i_0 k} \right\rangle + \sum_{j\neq i_0}\sum_{k\neq i_0} \left\langle \boldsymbol{P}_j\boldsymbol{P}_k^\intercal, \boldsymbol{X}_{jk}^* \right\rangle.$$

Since $\tilde{\boldsymbol{X}}_{ij} = \tilde{\boldsymbol{X}}_{ji}^\intercal$ and

$$\sum_{j\neq i_0} \left\langle \boldsymbol{P}_j\boldsymbol{P}_{i_0}^\intercal, \tilde{\boldsymbol{X}}_{ji_0} \right\rangle = \sum_{j\neq i_0} \left\langle (\boldsymbol{P}_j\boldsymbol{P}_{i_0}^\intercal)^\intercal, \tilde{\boldsymbol{X}}_{ji_0}^\intercal \right\rangle = \sum_{j\neq i_0} \left\langle \boldsymbol{P}_{i_0}\boldsymbol{P}_j^\intercal, \tilde{\boldsymbol{X}}_{i_0 j} \right\rangle,$$

$$\sum_{j\in[n]}\sum_{k\in[n]}\left\langle \boldsymbol{P}_j\boldsymbol{P}_k^\intercal,\tilde{\boldsymbol{X}}_{jk}\right\rangle=\left\langle \boldsymbol{P}_{i_0}\boldsymbol{P}_{i_0}^\intercal,\boldsymbol{I}_m\right\rangle+2\sum_{j\neq i_0}\left\langle \boldsymbol{P}_{i_0}\boldsymbol{P}_j^\intercal,\tilde{\boldsymbol{X}}_{i_0j}\right\rangle+\sum_{j\neq i_0}\sum_{k\neq i_0}\left\langle \boldsymbol{P}_j\boldsymbol{P}_k^\intercal,\boldsymbol{X}_{jk}^*\right\rangle$$

$$=-\left\langle \boldsymbol{P}_{i_0}\boldsymbol{P}_{i_0}^\intercal,\boldsymbol{I}_m\right\rangle+2\sum_{j\in[n]}\left\langle \boldsymbol{P}_{i_0}\boldsymbol{P}_j^\intercal,\tilde{\boldsymbol{X}}_{i_0j}\right\rangle+\sum_{j\neq i_0}\sum_{k\neq i_0}\left\langle \boldsymbol{P}_j\boldsymbol{P}_k^\intercal,\boldsymbol{X}_{jk}^*\right\rangle \qquad (52)$$

$$=-m+2\left(\left\langle \boldsymbol{P}_{i_0},\sum_{j\in[n]}\tilde{\boldsymbol{X}}_{i_0j}\boldsymbol{P}_j\right\rangle+\frac{1}{2}\sum_{j\neq i_0}\sum_{k\neq i_0}\left\langle \boldsymbol{P}_j\boldsymbol{P}_k^\intercal,\boldsymbol{X}_{jk}^*\right\rangle\right)$$

$$=-m+2\left(\left\langle \boldsymbol{P}_{i_0},\sum_{j\in[n]}\tilde{\boldsymbol{X}}_{i_0j}\boldsymbol{P}_j\right\rangle+\frac{1}{2}\sum_{j\neq i_0}\sum_{k\neq i_0}\left(m-\frac{1}{2}\left\|\boldsymbol{P}_j\boldsymbol{P}_k^\intercal-\boldsymbol{X}_{jk}^*\right\|_F^2\right)\right)$$

$$=C+2\left(\left\langle \boldsymbol{P}_{i_0},\sum_{j\in[n]}\tilde{\boldsymbol{X}}_{i_0j}\boldsymbol{P}_j\right\rangle-\frac{1}{4}\sum_{j\neq i_0}\sum_{k\neq i_0}\left\|\boldsymbol{P}_j\boldsymbol{P}_k^\intercal-\boldsymbol{X}_{jk}^*\right\|_F^2\right)$$

for some constant $C$. We note that the last term in the right hand side of (52) is a double sum of $(n-1)^2$ terms, which are independent of $i_0$. The minimization of this double sum over the variables $\{\boldsymbol{P}_j\}_{j\in[n]\setminus\{i_0\}}$ results in the ground-truth solution $\{\boldsymbol{P}_j^*\}_{[n]\setminus\{i_0\}}$ (since $jk\in E_g$ for $j,k\in[n]\setminus\{i_0\}$) with minimal value 0. Thus the right hand side of (52) can be viewed as a Langrangian with multiplier $\lambda=1/4$ of the constrained optimization problem

$$\max_{\{\boldsymbol{P}_i\}_{i\in[n]}\subset\mathscr{P}_m}\left\langle \boldsymbol{P}_{i_0},\sum_{j\in[n]}\tilde{\boldsymbol{X}}_{i_0j}\boldsymbol{P}_j\right\rangle \qquad (53)$$

$$\text{subject to}\quad \sum_{j\neq i_0}\sum_{k\neq i_0}\left\|\boldsymbol{P}_j\boldsymbol{P}_k^\intercal-\boldsymbol{X}_{jk}^*\right\|_F^2=0, \qquad (54)$$

which is equivalent to

$$\max_{\boldsymbol{P}_{i_0}\in\mathscr{P}_m}\left\langle \boldsymbol{P}_{i_0},\sum_{j\in[n]}\tilde{\boldsymbol{X}}_{i_0j}\boldsymbol{P}_j\right\rangle \qquad (55)$$

$$\text{subject to}\quad \boldsymbol{P}_j=\boldsymbol{P}_j^*\quad\text{for } j\neq i_0. \qquad (56)$$

We reformulate the above maximization problem by plugging its constraint into its objective function as follows:

$$\max_{\boldsymbol{P}_{i_0}\in\mathscr{P}_m}\left(\langle \boldsymbol{P}_{i_0},\boldsymbol{P}_{i_0}\rangle+\left\langle \boldsymbol{P}_{i_0},\sum_{j\neq i_0}\tilde{\boldsymbol{X}}_{i_0j}\boldsymbol{P}_j^*\right\rangle\right)=\max_{\boldsymbol{P}_{i_0}\in\mathscr{P}_m}\left(m+\left\langle \boldsymbol{P}_{i_0},\sum_{j\neq i_0}\tilde{\boldsymbol{X}}_{i_0j}\boldsymbol{P}_j^*\right\rangle\right). \qquad (57)$$

The above problem is almost similar to the one in the RHS of (47). They only differ in the term of the sum that correspond to $j=i_0$. Therefore, under the superspreader model, the least squares method is a regularized version of a similar energy function maximized on the RHS of (47). In a similar way to establishing (47), which results in wrongly estimating $\boldsymbol{P}_i^*$ as $\boldsymbol{P}_{\text{crpt}}$ by PPM, one can prove that under similar conditions to the ones of Proposition A.2 a least squares solver may produce $\boldsymbol{P}_{\text{crpt}}$ instead of $\boldsymbol{P}_{i_0}^*$.

## B   Additional Demonstration and Numerical Results

In §B.1 we provide a simple demonstration of the new idea in comparison to CEMP and IRLS. In §B.2 we briefly comment on the computational complexity of our methods. In §B.3 we present the experiments on a uniform corruption model. In §B.4 we provide additional results on the nonuniform corruption models.

### B.1   A Figure Demonstrating the IRGCL Algorithm

The following figure tries to convey the basic idea of IRGCL in comparison to IRLS and CEMP. In this figure, the notation $\boldsymbol{X}\rightarrow\boldsymbol{Y}$ means that $\boldsymbol{Y}$ is generated from $\boldsymbol{X}$. We recall that $\boldsymbol{A}$, $\boldsymbol{W}$, $\boldsymbol{P}$ and $\boldsymbol{S}^2$ respectively represent the estimated matrices of (correlation) affinity, weight, permutation and squared GCW. We also recall that $\boldsymbol{A}_1$ and $\boldsymbol{A}_2$ respectively denote the first and second order affinities. The two merged lines on the top of the diagram for IRGCL (one is dashed and the other is full) designate the fact that $\boldsymbol{A}$ is a weighted average of the first and second order affinities. We use a dashed line to remind the reader that the weights of $\boldsymbol{A}_1$ diminish as the number of iterations increases. We note that the two merged components represent two different algorithms, IRLS and CEMP.

Figure 2: Illustration of IRGCL and its relationship with CEMP and IRLS. The basic idea is that IRGCL is an iterative convex combination of CEMP and IRLS.

## B.2 On the Computational Complexity

We remark that the complexity of Algorithm 1 (which uses only 3-cycles) is $O(m^3 \times n^3)$. The complexity of the projected power iteration is $O(m^3 \times n^2)$. The spectral decomposition of the graph connection Laplacian has complexity $O(m^3 \times n^3)$. Thus, IRGCL-S&P, Spectral and PPM have the same complexity $O(m^3 \times n^3)$, which is typically lower than that of the SDP method MatchLift.

We remark that Algorithm 1 can be easily generalized to exploit higher order cycles with length $l$ by using the $l$-th power of the GCW matrix. In this case, its complexity is $O(m^3 \times n^3 \times l)$. On the other hand, the complexity of the original CEMP with general $l$-cycles is $O(m^3 \times n^l)$. Therefore, our idea significantly reduces the complexity of CEMP when using higher-order cycles and the specific metric discussed in this paper.

## B.3 Experiments on Uniform Corruption Model

We test the different methods using data generated from a uniform corruption model. In this model, we independently sample corrupted edges with probability $q$, and for each $ij \in E_b$, $\tilde{\boldsymbol{X}}_{ij} \sim \text{Haar}(\mathscr{P}_m)$.

We plot the estimation error

$$\sum_{i \neq j} \|\hat{\boldsymbol{X}}_{ij} - \boldsymbol{X}_{ij}^*\|_F^2 / \sum_{i \neq j} \|\boldsymbol{X}_{ij}^*\|_F^2$$

for each corruption probability $q = 0.7, 0.8, 0.88, 0.9$ and $0.92$. We compare IRGCL-P and IRGCL-S with all methods described in §6. Since IRLS-Cauchy-S and IRLS-Cauchy-P performed similarly we report only one of them. We also tested the standard IRLS described in (5) and (6), which we refer to as IRLS-L1-S. The implementation of IRLS-L1-S approximately solves (6) using the spectral formulation of (12) at each iteration. It also initializes by the solution of (12) using the adjacency matrix for the weight matrix. For each method we run 100 trials and report the means and standard deviations of the estimation errors in Figure 3, where standard deviations are denoted by error bars. We note that IRGCL-S and IRGCL-P consistently

Figure 3: Average matching error under a uniform corruption model.

achieve the lowest errors, and IRGCL-S seems to work slightly better (with lower mean errors and standard

deviations) under the highest corruption ratio, $q = 0.92$. The spectral method and MatchLift perform the worst. They are unable to recover the ground-truth permutations when $q = 0.8$. We also remark that PPM works better than the other least squares methods. However, it is not competitive with IRGCL-S and IRGCL-P in the high corruption range of $0.88 - 0.92$. In this range, IRLS-L1-S and IRLS-Cauchy-S have lower means than PPM, but they have large standard deviations, which indicate that they are unstable.

## B.4    Additional Experiments on Nonuniform Corruption Models

We report additional results for the LBC and LAC models in §B.4.1 and §B.4.2 respectively. Numerical results for an Erdős-Rényi graph are included in §B.4.3.

### B.4.1    Additional synthetic experiments using the LBC model

Figure 4 reports the estimation errors
$$\sum_{ij \in E_b} \|\hat{X}_{ij} - X^*_{ij}\|^2_F / \sum_{ij \in E_b} \|X^*_{ij}\|^2_F$$
of different methods under the LBC model with parameters $m_c = 90$ and $n_c = 10, 20, 30, 40$. For each method and each fixed value of $n_c$ we run 20 trials and present the mean and standard deviations of the estimation errors. We note that both IRGCL-S and IRGCL-P are able to achieve near exact recovery when $n_c \le 30$. PPM

Figure 4: Average matching errors under the local biased corruption model.

performs the worst among the tested methods for all values of $n_c$. We note that in terms of the averaged errors, IRLS-Cachy-S performs better than the other least squares methods. However, it has high standard deviations, so that it is unstable, and its averaged values are still not competitive when compared with IRGCL-S and IRGCL-P. We also note that the standard deviations of the latter two methods are nearly 0 when $n_c \le 30$.

### B.4.2    Additional synthetic experiments using the LAC Model

Figure 5 reports the estimation errors
$$\sum_{ij \in E_b} \|\hat{X}_{ij} - X^*_{ij}\|^2_F / \sum_{ij \in E_b} \|X^*_{ij}\|^2_F$$
of different methods under the LAC model with $m_c = 60$ and $n_c = 10, 20, 30, 40$. For each method and each value of $n_c$ we run 20 trials and report the mean and standard deviations of the errors. We note that both IRGCL-P and IRGCL-S are able to recover the ground-truth solution under the LAC model when $n_c \le 40$, whereas other methods cannot.

### B.4.3    Additional synthetic experiments with an Erdős-Rényi graph

We repeat the experiments in the main text with $G([n], E)$ as an Erdos-Renyi graph with probability 0.5 instead of a complete graph. Figure 6 reports the estimation errors
$$\sum_{ij \in E_b} \|\hat{X}_{ij} - X^*_{ij}\|^2_F / \sum_{ij \in E_b} \|X^*_{ij}\|^2_F$$

Figure 5: Average matching errors under the local adversarial corruption Model.

of different methods under the LAC model with $m_c = 30$ and LBC model with $m_c = 45$. Both models have $n_c = 1, 2, 3, 4, 5, 6$. For each method and each value of $n_c$ we run 20 trials and report the mean and standard deviations of the errors. We also report the final error of IRGCL-S and IRGCL-P compared with $P_{(1)}$ in Algorithm 2 (we call it IRGCL-init) in figure 7.

We note that both IRGCL-P and IRGCL-S are able to give exact recovery on LAC and almost exact recovery on LBC, while other methods cannot. Also, we find that on LBC both IRGCL iterations effectively decrease error compared to its initialized permutation, though the initialization is already quite good in this synthetic setting; On LAC the initialization of IRGCL already achieves exact recovery.

Figure 6: Average matching errors under the local biased corruption model (left) and local adversarial corruption model (right) with an Erdős-Rényi graph with $p = 0.5$.

Figure 7: Average matching errors of IRGCL-S and IRGCL-P compared with IRGCL initialization under the local biased corruption model (left) and local adversarial corruption model (right) with an Erdős-Rényi graph with $p = 0.5$.