[Reviews · NeurIPS 2020]

Review 1

Summary and Contributions: The paper investigates multi-object matching via robust permutation synchronization. The authors propose an iterative reweighting strategy, where the weights are initialized using CEMP (a technique borrowed from related work, but with a novel interpretation) and then an iterative approach alternates weight update and permutation estimation. Performance guarantees under a specific corruption model are proven. The approach is also validated in simulated and real datasets.

Strengths: - The paper tackles an interesting problem. - The discussion of the limitations of IRLS methods is reasonable and highlights shortcomings of existing methods. - The theorems, drawing connections with CEMP, are interesting and sound. Moreover, the performance guarantees in Theorem 5.2 were very well received. It is indeed important to understand the theoretical limits of robust synchronization algorithms. - The approach works very well in practice and dominates baseline methods, as can be seen in Fig. 1 and Table I.

Weaknesses: - The presentation can be largely improved: some of the sentences do not help the reader and currently the draft is not easy to read. For instance, the transition to the second paragraph of the introduction is quite abrupt: while the first paragraph talks about keypoints and images, the keypoints disappear when talking about the permutation synchronization. Moreover, the sigma is not properly defined. Similarly, Section 4.1 is hard to read: while the overall idea is clear, the reader gets trapped in too many details, such as those in lines 196-199. Similar comments hold for Section 4.2: why presenting two techniques when you recommend using only one? What about moving the other to the appendix? While I understand the desire of the authors to be comprehensive, mentioning too many details may compromise clarity. - The literature review misses several related papers on permutation synchronization: [1] Bernard, Thunberg, Goncalves, Theobalt, Synchronisation of partial multi-matchings via non-negative factorisations, Pattern Recognition, 2019. [2] Serlin, Sookraj, Belta, Tron. Consistent Multi-Robot Object Matching via QuickMatch. In International Symposium on Experimental Robotics, 2018. [3] Fathian, Khosoussi, Tian, Lusk, How, CLEAR: A Consistent Lifting, Embedding, and Alignment Rectification Algorithm for Multi-View Data Association, Arxiv, 2019. As well as other papers on synchronization on Lie groups: [4] Rosen, Carlone, Bandeira, Leonard, A certifiably correct algorithm for synchronization over the special Euclidean group, IJRR, 2019. [5] Arrigoni, Fusiello, Synchronization problems in computer vision with closed-form solutions, IJCV, 2020. - Section 5: It is unclear why the superspreader model (in particular, in conjunction with assumption (12)) is more realistic or more challenging than the uniform corruption. It might be good to add more comments. Also: why restricting to a complete graph? It seems unrealistic for the SfM (and other) applications mentioned in the intro. - The approach (and Algorithm 2) is technically sound but looks overly complicated. If CEMP can reliably estimate the corruption levels, isn’t it enough to do a single weighted least squares optimization afterwards (ie stopping before the "for" loop in Algorithm 2)? am I missing something? Indeed Theorem 5.2 seems to guarantee the performance of CEMP rather than the overall IRGCL algorithm. - Related to the previous point: it would be good to assess the performance of CEMP in the experiments. In other words, how good is P_(1) in Algorithm 2? - Section 6.2 preprocess the real dataset to make it usable by the proposed algorithm (e.g., by removing some of the corrupted measurements): how would that work in practice?

Correctness: The technical results seem correct. The empirical evaluation can be improved since it requires pre-processing the real datasets, which seems unrealistic.

Clarity: The presentation is the main issue of the paper. I think the technical contribution is nice, but the presentation is hard to follow.

Relation to Prior Work: The literature review can be improved by adding missing references (see previous comments) and better discussing the relations with CEMP.

Reproducibility: Yes

Additional Feedback: - line 39: “the the” - line 213: “it faster” -> “it is faster” - Line 175: “CEMP is provably robust to adversarial corruption, but is more difficult to motivate”: this sentence is confusing and CEMP is not well reviewed in the related work section. - Line 99: sigma_ij should have a tilde I think - It’s the measured quantity. POST REBUTTAL: I found the rebuttal very convincing. I agree with R2 that adding more experiments (and discussing computational aspects) could further improve this paper. R3's concern is mainly about discussing connections with IRLS (which seems fixable) and comments on convergence -- as far as I see, the authors' rebuttal provides fairly convincing comments. In general, having a IRLS-like algorithm that can tackle permutation synchronization seems interesting and novel to me. I also agree with R4 about the lack of clarity: that seems to be the main issue of the currrent paper. In summary, I like the paper and I'm convinced about the technical quality, but the presentation is not very accessible and I hope the authors can improve in the final submission.


Review 2

Summary and Contributions: The main contribution of this work is an iterative reweighting algorithm for the permutation synchronization problem. The authors do a good job at placing the work in the context of the group synchronization problem, of recovering (compact) group elements from a small noisy subset of pairwise measurements. The link between the two is given by the standard representation of the permutation group elements as doubly stochastic binary matrices.

Strengths: The paper addresses a rather niche regime (of heterogeneity) in which other methods do not seem to perform well. The obvious previous work on the topic of this paper is [22] D. Pachauri, R. Kondor, and V. Singh. Solving the multi-way matching problem by permutation synchronization. NIPS 2013 which essentially leveraged the group synchronization existing literature at the time to propose a spectral methods for permutation synchronization. The popular iteratively reweighted least squares (IRLS) does not trivially extend itself applicable to the permutation synchronization problem due to the discrete nature of the permutations which may lead to zero residuals, which in turn will lead to overweighted edges in the iterative scheme. The authors proposed an interesting scheme that involved iteratively reweighing edges in the graph Connection Laplacian, and draws similarities with the popular VDM framework of Singer and Wu, that extends the classical diffusion maps framework to incorporate not only scalar similarities between the nodes, but also the orthogonal transformations between bases for the tangent places at points close enough on the manifold. The same matrix operator is leveraged here, with additional normalizations and weighting schemes. The authors provide performance guarantees under the adversarial effect of nonuniform corruption, with bad/corrupted edges being attached to a single node in a star shaped topology. I think this model is very realistic one and often encountered but understudied in the literature, and a nice contribution overall. (For example in certain graph embedding problems, where nodes represent embedding of overlapping subgraphs, and edges hold pairwise ratios of the group element at the endpoints of each edge. If a subgraph embedding is grossly corrupted, so will be the information contained on the incident edges). I think this is an interesting paper. I could see one argue that the novelty is slightly incremental compared with the approaches in works such as [22] and the given VDM framework, but I think the idea of reweighing in this context is neat and well motivated by the fact that existing methods such as [22] only cope well with uniform corruption which is less realistic in real applications. It would be very interesting to extend this to the setting of group synchronization over other groups, if this is something that has not been looked at yet. Overall, I think this paper could be of impact in the computer vision literature.

Weaknesses: The authors should perhaps also show simulations when the measurement graph is not complete, but also vary the sparsity level, in addition to the noise levels. It would be interesting to see how the different methods compare in this regime (below (log n)/n), especially since spectral methods often require regularization to handle such sparsity. Can the authors also comment on the computational aspects? Especially given the data sets considered, in both synthetic and real data, are fairly limited in size.

Correctness: The paper appears to be technically sound.

Clarity: Yes, the paper is well written and motivated. The authors could spend some in the intro to briefly explain the group synchronization problem to readers less familiar with this literature.

Relation to Prior Work: Yes, the authors explain how this work relates to existing literature and what gaps it aims to fill.

Reproducibility: No

Additional Feedback: ----- Post Rebuttal ----- After reading the rebuttal, I am happy to maintain my initial score and think this is a solid submission. The authors should address the few loose ends for the final version.


Review 3

Summary and Contributions: The paper presents a method for multi graph matching that exploits detailed more detailed problem constraints than existing methods.

Strengths: Advances the state-of-the art showing a way to solve a relevant and complex problem with a more sophisticated approach than existing methods.

Weaknesses: One aspect of the papers that seems off-putting are the very strong wording regarding IRLS, without offering much insight in the ways the proposal differs from IRLS. This is made more confusing by the fact the proposed algorithm does seem to follow in the end a recipe similar to IRLS, with a weighted least-squares problem solved iteratively with the proposed method basically attaining itself to how the weights are calculated. Is it really the case that the proposed method cannot be understood as IRLS, although perhaps with a weight calculation that is much more sophisticated that previous research has ever done? The proposed method seems to be carefully exploiting many relevant constraints of the problem, and achieving great results from that, what is all great. Is it really fundamentally contradicting the (very general) IRLS algorithm, though? It's important to understand how the method relates to IRLS because it's a great theoretical framework even before being a very convenient, practical method. Two important questions show up when IRLS is applied to any problem in general: 1_ what is the corresponding non-linear error function that will be minimized as a consequence of the formula utilized to calculate the weights? 2_ Is there any probabilistic interpretation of the process, viewing it as a case of Expectation-Maximization? Other important considerations are: how important is initialization to the process? The authors did seem to touch this issue a bit, although their understanding does not seem to be very clear. While IRLS is not exactly a non-linear optimization method such as Gauss-Newton, Levenberg-Marquadt, etc, it's quite safe to say it falls it that family. Meaning it mostly will provide you a local minimum from the attraction basin of the initial solution. That means it's not a global optimization method, and therefore a "bad" initialization along with an objective function that contains many sub-optimal local minima will be a more challenging problem to solve. The paper mentions this local minima "problem" with IRLS. First of all, it should be clear this is not a problem with the optimization method, really. The real problem is with the objective function and the initialization. The problem is initializing out of the optimal basin. And of course, anyone would prefer a global optimization if it were feasible. What is the case with the proposed method? Is it a global optimization method, unlike IRLS, Gauss-Newton, LM, etc? Or is it a local optimization method, except the objective function proposed by the authors tends to have more convenient attraction basins, and the initialization proposed tends to lead to a global optimum, or at least a very good one? These are very important details about how such algorithm works, and while the authors bring some of these concepts relative to the alternatives, it's not so clear what happens in the proposal. Could it not be the case that the proposal is still pretty much IRLS, except the non-linear error being utilized has vastly advantageous properties? The paper right now even seems to suggest that the proposed method is actually performing some form of global optimization, and it would be very nice to make it clear what is the case.

Correctness: Experiments seem adequate.

Clarity: It could be improved in terms of clarity and accessibility to a wider audience. The paper begins by citing very concrete problems such as SfM, and later attains to abstract and specific concepts that only someone knowledgeable of the related methods could grasp. It is hard to understand how practical concepts such as reprojection errors or descriptor distances get translated into graph weights. This is probably well-understood by anyone aware of the methods, although some pedagogy would be appreciated. This is important because these specifics might make a difference in the performance of the method, and why it can outperform simpler alternatives.

Relation to Prior Work: There is plenty of comparisons with previous methods.

Reproducibility: Yes

Additional Feedback:


Review 4

Summary and Contributions: This paper describes a new solution for multiple object matching (permutation synchronization) problem, based on an iteratively reweighting scheme. The solution outperforms the conventional IRLS approach thanks to the superior handling of non-uniform noises presented in the input.

Strengths: The method is shown to perform well, and better than conventional IRLS or least square method for permutation synchronization. A theoretic proof is provided that assures the robustness of the method in the presence of non-uniformly distribution corruptions. The central idea of the method can be extending to solving other multi-object "group synchronization" tasks, and partial matching problems in 3D reconstruction.

Weaknesses: Despite the task of multi-object permutation synchronization itself is conceptually simple and (should be) easy to understand, the paper is written in a form making it unnecessarily difficult to follow. Many of the writing styles and notational systems are odd and unintuitive. Although I consider I under stand most parts of the paper, From time to time I had to refer back and forth in the paper the notations due to their unnatural choices of symbols. In other parts, I felt at lost, e.g. why do you require the relative permutation "\sigma^*_{ij} = \sigma^{*-1}_{ij}" rather than a commutative form ? I cannot see where you give the full term for CEMP other than its reference [17]. These issues are in my view not merely language issues, or poor writing. They reflect the authors may have not spent sufficient time to organize their thoughts and the paper's structure well. The description of the graphical structure used in the paper is also unclear. Was it a complete graph or a singly connected cycle graph ? In the context of SFM (e.g [29]) the graph structure is determined by the visibilities of the multiple input camera views. More experiments are needed on this aspect in order to evaluate the effects of different graph connections. In general, I find the experiment section is thin and fail to valid many of the claims (of advantages) that the paper makes in the main text of the paper.

Correctness: Appears so, though have not checked completely.

Clarity: Not satisfactory... and can be substantially improved. See comments above.

Relation to Prior Work: The current work seems to draw inspiration from multiple prior works. The literature review section (1.1) lacks in depth analysis and comparison. It could have been better written , to place the contribution of this work in context.

Reproducibility: No

Additional Feedback:

[Author Response · NeurIPS 2020]

We thank the reviewers for the valuable comments, which require simple changes to the manuscript.

**R1**: 1) We can easily address the comments on the writing, while pushing some unnecessary technical details to the supplementary material. We remark that we had a typo in line 86 and should instead have $\sigma_{ij}^* = \sigma_{ji}^{*-1}$. 2) We will include the mentioned references. 3a) Condition (12) in our model can be interpreted as follows: in a 3-cycle, corrupting two edges induces on average more cycle-inconsistency than corrupting a single edge. This is currently explained in line 503 of the supp. material (also note our correction in line 614). Corruption with a uniform (Haar) distribution also satisfies this condition (with equality), however, it has a very strong and unrealistic assumption on the distribution. For simplicity we have a star-shaped topology of our graph, though with more work (and rather complicated descriptions) one may generalize the topology of our graph. We emphasize departure from the uniformity assumption on the distribution of corruption and not the "uniform topology" of the graph. 3b) Although our theory and experiments assume a complete graph, for simplicity of demonstration, they can be easily generalized to the incomplete Erdős-Rényi graphs. We will add this to the supp. material. 4) We remark that IRGCL outperforms CEMP+weighted least squres (WLS), which reports $\boldsymbol{P}_{(1)}$, in highly corrupted scenarios. For example, under the uniform corruption model with $n=100$ and $90\%$ corrupted edges, the error of IRGCL is $<0.1$, while $\boldsymbol{P}_{(1)}$ gives estimation error $>0.5$ (like other methods reported in Figure 3). That is, given poor initialization, IRGCL can still converge to a reasonable solution. We believe that IRGCL uses additional cycle consistency information to adjust the edge weights (see also reply to R3). However, under mild corruption, the CEMP initialization is accurate and helpful in accelerating the convergence and this is what the theory verifies (previous theory of [17] for high uniform corruption requires very large $n$). 5) We will report the result of CEMP+WLS (that is, using $\boldsymbol{P}_{(1)}$) in the updated version. 6) The original real data used in our work is the most challenging one for permutation synchronization. However, it contains many nodes whose neighboring edges are completely corrupted. In such a case, none of the permutation synchronization algorithms work well and additional information, such as coordinates of key points, is needed. Thus, in order to make a valid evaluation of different algorithms, we have to preprocess this data so that permutation synchronization is well-posed. We remark that in many SfM data (e.g. the initial matching used in [20]) such a malicious scenario does not occur (thus no such preprocessing is needed); however, currently our algorithm and other direct algorithms for permutation synchronization cannot be easily applied to SfM data since they deal with permutations and not partial ones. The nontrivial extension to partial permutations is left for future work.

**R2**: 1). We will demonstrate this in practice. In theory, for 3-cycles and uniform corruption, a necessary condition for CEMP and IRGCL is $p=\Omega(1/\sqrt{n})$. According to [17], up to a log factor this condition is sufficient for CEMP. As the length of the cycle increases the lower bound on $p$ decreases and approaches $p=\Omega(\log n/n)$ as the size of the cycle approaches infinity. The complexity of our reformulation of CEMP is at most $n^3$ times the length of the cycle, so in practice we cannot achieve the $\log n/n$ threshold of disconnectivity, but get close to it. 2) We will report runtimes in the new version. We remark that IRGCL is often slightly slower than IRLS, but they are comparable to each other. Our experiments indicate the following order of runtimes: Spectral<PPM<IRLS<IRGCL≪MatchALS<MatchLift.

**R3**: The challenge of permutation synchronization is not just its nonconvexity, but more importantly, its discrete and combinatorial nature. IRLS has been carefully studied and tested in some continuous settings, but in discrete settings IRLS is neither commonly applied nor studied. Indeed, in lines 152-160, we explain the drawback of IRLS in our discrete setting. IRGCL handles the limitations of IRLS in the following ways. 1) Recall that IRLS first locally estimates a residual based on a single measurement of the corresponding edge. However, IRGCL estimates the residual using more global information of other edges (reflected in the powers of the GCW matrix). Thus its WLS is much less dependent on the initialization than standard IRLS. Point 4) in our reply to R1 confirms that in practice IRGCL can handle cases where CEMP provides a bad initialization. 2) The solution of the IRLS problem uses convex relaxation of the nonconvex WLS problem. While IRGCL also follows such a scheme, it also uses 3-cycle consistency information which helps more faithfully recover the underlying corruption and thus provide more accurate weights and consequently a better approximation by the convex relaxation. 3) Our edge weights are computed as a weighted average (ideally, expectation) of the 3-cycle consistency (encoded in the square of the GCW matrix). This expectation lies in a continuous space (as oppose to the weights in IRLS that lie in a discrete space). Thus our reweighting scheme smooths the space of edge weights, making the algorithm less likely to get stuck. We remark that CEMP does not explicitly minimize an objective function, but it aims to find the underlying maximal cycle-consistent subgraph (see page 10 of [17]). It can also be interpreted as an iterative procedure that aims to estimate the expectation of the corruption level of edges (latent variables) given their posterior distribution, which is similar to EM. However, its iterations do not rely on MLE and are more efficient and thus have some nice guarantees of convergence (see [17]). Similarly, IRGCL aims to solve a WLS problem whose weights focus on the underlying globally cycle-consistent subgraph. In order to do this, it estimates both permutations (using WLS) and the cycle-consistent subgraph (using CEMP-like reweighting) in an alternating manner.

**R4**: This review is an outlier in terms of short length, tone, clarity and score. As explained to R3, our work has some nonstandard ideas and we strived to make it accessible, while avoiding some of the complicated ideas of [17] (and it is nice that we have a direct formulation by the connection graph). We will follow all the constructive suggestions and easily improve the clarity, but we disagree that we did not organize our thoughts. The answer to all your points are above, except for the following: 1) The graph is not "a singly connected cycle graph" (and we never stated this). 2) Most previous works on permutation synchronization use real datasets whose images have similar views and share the same set of keypoints (so that keypoint matches are permutations) and their graphs are thus complete. Applying permutation synchronization algorithms to a set of images with distinct views is unrealistic and is not a common practice (the reviewer may be confused with rotation synchronization). Future work will try to generalize our ideas to partial permutations so that we may explore more general SfM data.

[Meta-Review · NeurIPS 2020]

This paper generated quite a bit of discussion. While three of the reviewers found the paper to be above the acceptance threshold, the fourth one was concerned about the clarity of the writing and the relationship to IRLS. In the rebuttal, the authors called R4's review a "an outlier" but did a good job of responding to comments from the other three reviewers. I am happy to recommend accepting this paper but ask that the authors prepare a significantly revised version that addresses the issues raised by the reviewers. I should point out that the concerns raised by R4 were supported by the other reviewers as well during discussion, hence he/she is not an outlier.